# Social and environmental predictors of gut microbiome age in wild baboons

Mauna R Dasari[1,2,3]*, Kimberly E Roche[4], David Jansen[1], Jordan Anderson[5], Susan C Alberts[5,6,7], Jenny Tung[5,6,7,8,9,10], Jack A Gilbert[11], Ran Blekhman[12], Sayan Mukherjee[13,14,15], Elizabeth A Archie[1]*

[1]Department of Biological Sciences, University of Notre Dame, Notre Dame, United States; [2]Department of Biological Sciences, University of Pittsburgh, Pittsburgh, United States; [3]California Academy of Sciences, San Francisco, United States; [4]Program in Computational Biology and Bioinformatics, Duke University, Durham, United States; [5]Department of Evolutionary Anthropology, Duke University, Durham, United States; [6]Department of Biology, Duke University, Durham, United States; [7]Duke University Population Research Institute, Duke University, Durham, United States; [8]Department of Primate Behavior and Evolution, Max Planck Institute for Evolutionary Anthropology, Leipzig, Germany; [9]Canadian Institute for Advanced Research, Toronto, Canada; [10]Faculty of Life Sciences, Institute of Biology, Leipzig University, Leipzig, Germany; [11]Department of Pediatrics and the Scripps Institution of Oceanography, University of California, San Diego, San Diego, United States; [12]Section of Genetic Medicine, Department of Medicine, University of Chicago, Chicago, United States; [13]Departments of Statistical Science, Mathematics, Computer Science, and Bioinformatics and Biostatistics, Duke University, Durham, United States; [14]Center for Scalable Data Analytics and Artificial Intelligence, University of Leipzig, Leipzig, Germany; [15]Max Planck Institute for Mathematics in the Natural Sciences, Leipzig, Germany

*For correspondence:
mauna.dasari@gmail.com (MRD);
earchie@nd.edu (EAA)

## eLife Assessment

This study leverages an impressive and comprehensive longitudinal 16S rRNA gut microbiome dataset from baboons to provide **important** insight regarding the use of a microbiome-based clock to predict biological age. The evidence for age-associated microbiome features and environmental and social variables that impact microbiome aging is **convincing**. This study of microbiomes as markers of host age will fuel inquiries and studies and interest a broad range of researchers, especially those interested in alternatives to measuring biological aging.

**Abstract** Mammalian gut microbiomes are highly dynamic communities that shape and are shaped by host aging, including age-related changes to host immunity, metabolism, and behavior. As such, gut microbial composition may provide valuable information on host biological age. Here, we test this idea by creating a microbiome-based age predictor using 13,563 gut microbial profiles from 479 wild baboons collected over 14 years. The resulting 'microbiome clock' predicts host chronological age. Deviations from the clock's predictions are linked to some demographic and socio-environmental factors that predict baboon health and survival: animals who appear old-for-age tend to be male, sampled in the dry season (for females), and have high social status (both sexes). However, an individual's 'microbiome age' does not predict the attainment of developmental milestones or lifespan. Hence, in our host population, gut microbiome age largely reflects current,

as opposed to past, social and environmental conditions, and does not predict the pace of host development or host mortality risk. We add to a growing understanding of how age is reflected in different host phenotypes and what forces modify biological age in primates.

## Introduction

For most vertebrate species, physical declines with age are inevitable. These changes define the concept of biological aging and contribute to increased disease burden in older individuals (*Komanduri et al., 2019*; *López-Otín et al., 2013*). While vertebrates have species-typical patterns of biological aging, those patterns can also differ between individuals within species. Hence, an animal's age in years—that is, its chronological age—is not an exact reflection of age-related decline in physical functioning (*Nakamura and Miyao, 2007*; *Gems and Partridge, 2013*; *Belsky et al., 2015*; *Hayward et al., 2015*). Measuring individual differences in biological age is an important first step to understand how socio-environmental conditions influence aging processes and to identify strategies to improve health in old age.

In mammals, one valuable marker of biological aging may lie in the composition and dynamics of the gut microbiome (*Claesson et al., 2012*; *Heintz and Mair, 2014*; *O'Toole and Jeffery, 2015*; *Sadoughi et al., 2022*). Age-related changes in gut microbiomes are well documented in humans and other animals, and the gut microbiome has the potential to reflect a wide variety of aging processes for individual hosts (*Bäckhed et al., 2005*; *Mueller et al., 2006*; *Koenig et al., 2011*; *Yatsunenko et al., 2012*; *Bergström et al., 2014*; *Langille et al., 2014*; *Clark et al., 2015*; *Cong et al., 2016*; *Yassour et al., 2016*; *Biagi et al., 2016*; *Odamaki et al., 2016*; *Smith et al., 2017*; *Reese et al., 2021*; *Baniel et al., 2021*). Mammalian gut microbiomes interact with the immune, endocrine, nervous, and digestive systems, all of which change with age (*Bäckhed et al., 2012*; *Foster et al., 2017*; *Clayton et al., 2018*; *Martin et al., 2019*). Gut microbiomes are also sensitive to host environments and behaviors that change with age, including host diet, living conditions, and social integration (*Reese et al., 2021*; *Bengmark, 1998*; *Claesson et al., 2011*; *Gerber, 2014*; *Palmer et al., 2007*). Finally, gut microbiomes may play a causal role in age-related changes in host development and longevity and may, therefore, be directly involved in individual differences in biological age (*Langille et al., 2014*; *Clark et al., 2015*; *Smith et al., 2017*; *Salosensaari et al., 2021*; *Wilmanski et al., 2021*). For example, children who experience famine exhibit developmentally immature gut microbiomes that, when transplanted into mice, delay growth and alter bone morphology (*Smith et al., 2013*; *Subramanian et al., 2014*; *Blanton et al., 2016*; *Gehrig et al., 2019*). Experiments in flies, mice, and killifish find that the gut microbiome can also influence longevity (*Langille et al., 2014*; *Clark et al., 2015*; *Smith et al., 2017*; *Tian et al., 2017*).

One strategy for testing if gut microbiomes reflect host biological age is to apply supervised machine learning to microbiome compositional data to develop a model for predicting host chronological age, and then to test whether deviations from the resulting 'microbiome clock' age predictions are explained by socio-environmental drivers of biological age and/or predict host development or mortality. A parallel approach is commonly applied to patterns of DNA methylation, and epigenetic clock age estimates have been shown to predict disease and mortality risk more accurately than chronological age alone (*Horvath, 2013*; *Marioni et al., 2015*; *Chen et al., 2016*; *Binder et al., 2018*; *Declerck and Vanden Berghe, 2018*; *Anderson et al., 2021*). To date, at least five microbiome age-predicting clocks have been built for humans, which predict sample-specific age with median error of 6–11 years (*de la Cuesta-Zuluaga et al., 2019*; *Galkin et al., 2020*; *Huang et al., 2020*; *Chen et al., 2022*). However, to our knowledge, no clocks have tested whether microbiome age is sensitive to potential socio-environmental drivers of biological age or predicts host development or mortality.

Here, we create a microbiome-based age-predicting clock using 13,476 16S rRNA gene sequencing-based gut microbiome compositional profiles from 479 known-age, wild baboons (*Papio* sp.) sampled over a 14-year period (*Figure 1A, B*). These microbiome profiles represent a subset of a dataset previously described in *Grieneisen et al., 2021*, *Björk et al., 2022*, and *Roche et al., 2023*, filtered to include only the baboon hosts whose ages were known precisely (within a few days' error). Important to human aging, baboons share many developmental similarities with humans, including an extended juvenile period, followed by sexual maturation and non-seasonal breeding across adulthood

**eLife digest** As we age, our bodies undergo a variety of physical changes. However, the pace at which these changes occur (known as our biological age) often does not reflect the number of years we've lived (known as our chronological age).

Various markers have been proposed to predict biological age, including the composition of bacteria living in the gut. Which bacterial species reside in the gut is influenced by multiple factors, such as diet, living conditions and social interactions. This makes the microbiome unique to each individual, and potentially a rich indicator of age-related processes.

To explore this idea, Dasari et al. studied a large dataset containing thousands of gut microbiome samples from almost 500 wild baboons, collected over 14 years. Several machine learning algorithms were applied to the data to estimate the 'microbiome age' of each individual. Dasari et al. found that these estimates correlated well with the baboons' chronological ages, and mirrored known patterns of biological aging, such as male baboons aging faster than females.

Environmental and social factors – such as a baboon's social rank within a group – also influenced the relationship between chronological and biological age. During the dry season, for instance, female baboons had a higher microbiome age compared to their actual age, and baboons with low social status had a lower microbiome age than expected.

Although life expectancy has steadily increased over the last century, our healthspan (the period of life spent in good health) has not kept pace with it. Understanding how our bodies age is key to prolonging healthspan. The findings of Dasari et al. suggest that the gut microbiome is a good predictor of biological age, and future work investigating this relationship could provide valuable clues for slowing down the aging process.

(*Figure 1C, D*; *Björk et al., 2022*; *Roche et al., 2023*; *Alberts and Altmann, 1995*; *Altmann et al., 2010*).

The baboons in our dataset were members of the well-studied Amboseli baboon population in Kenya, which has continuous, individual-based data on early-life environments, maturational milestones, social relationships, and mortality (*Alberts et al., 2014*; *Archie et al., 2014a*; *Archie et al., 2014b*; *Grieneisen et al., 2017*; *Ren et al., 2016*; *Tung et al., 2015*). Relevant to measuring biological age, prior research in Amboseli has identified several demographic, environmental, and social conditions that predict physical condition, the timing of development, or survival, including sex, season, social status (i.e., dominance rank), and early-life adversity (*Alberts and Altmann, 1995*; *Altmann et al., 2010*; *Charpentier et al., 2008*; *Bronikowski et al., 2011*; *Gesquiere et al., 2011*; *Archie et al., 2012*; *Lea et al., 2015*; *Tung et al., 2016*; *Gesquiere et al., 2018*; *Zipple et al., 2019*). Consistent with the possibility that these associations arise from causal effects of harsh or stressful conditions on biological aging (*Jovanovic et al., 2017*; *Zannas et al., 2015*; *Raffington et al., 2020*), and with the idea that microbiomes serve as a marker of biological age (*Claesson et al., 2012*; *Heintz and Mair, 2014*; *O'Toole and Jeffery, 2015*; *Sadoughi et al., 2022*), we tested whether socio-environmental conditions predicted microbiome age estimates. Our predictions about the direction of these effects varied depending on the socio-environmental condition and the developmental stage of the animal (i.e., juvenile or adult).

In terms of sex, adult male baboons exhibit higher mortality than adult females. Hence, we predicted that adult male baboons would exhibit gut microbiomes that are old-for-age, compared to adult females (by contrast, we expected no sex effects on microbiome age in juvenile baboons).

In terms of season, the Amboseli ecosystem is a semi-arid savannah with a 5-month long dry season during which little rain falls, often leading to nutritional hardship (*Alberts and Altmann, 2012*). We predicted that samples from the dry season might appear to be old-for-age, compared to samples from the wet season due to nutritional stress in this difficult season.

In terms of social status, baboons experience linear, sex-specific hierarchies. Female ranks are nepotistic, with little social mobility, and low rank is linked to low priority of access to food (*Charpentier et al., 2008*; *Gesquiere et al., 2018*; *Melnick and Pearl, 1987*; *Altmann and Alberts, 2005*; *Silk et al., 2003*). In contrast, adult male rank is determined by strength and fighting ability and is dynamic across adulthood (*Alberts et al., 2003*). High-ranking males experience high energetic costs

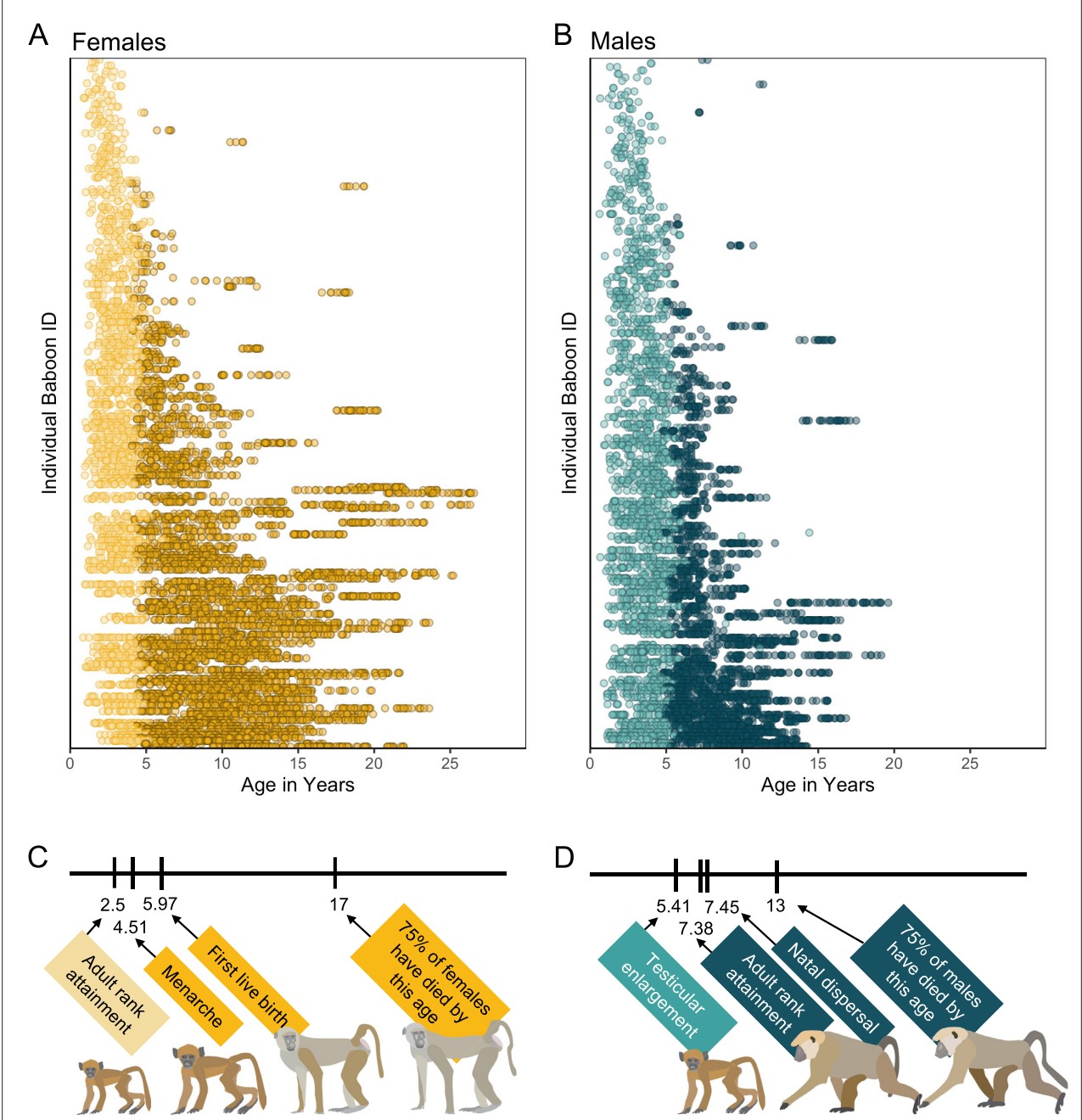

**Figure 1.** Microbiome sampling time series and developmental milestones for the Amboseli baboons. Plot (**A**) shows microbiome samples from female baboons, and plot (**B**) shows samples from male baboons. Each point represents a microbiome sample from an individual subject (*y*-axis) collected at a given host age in years (*x*-axis; *n* = 8245 samples from 234 females shown in yellow; *n* = 5231 samples from 197 males shown in blue). Light and dark point colors indicate whether the baboon was sexually mature at the time of sampling, with lighter colors reflecting samples collected prior to menarche for females (*n* = 2016 samples) and prior to testicular enlargement for males (*n* = 2399 samples). Due to natal dispersal in males, we have fewer samples after the median age of first dispersal in males (*n* = 1705 samples, 12.6% of dataset) than from females after the same age (*n* = 4408 samples, 32.6% of dataset). The timelines below the plots indicate the median age in years at which (**C**) female baboons attain the developmental milestones analyzed in this paper—adult rank, menarche and first live birth—and (**D**) males attain adult rank, testicular enlargement, and disperse from their natal groups (*Charpentier et al., 2008*; *Onyango et al., 2013*). The age at which 75% of animals in the population have died is shown to indicate different life expectancies for females versus males (*Bronikowski et al., 2011*). Baboon illustrations courtesy of Emily (Lee) Nonnamaker.

of mating effort have altered immune responses, and exhibit old-for-age epigenetic age estimates compared to low-ranking males (*Anderson et al., 2021*; *Gesquiere et al., 2011*; *Archie et al., 2012*). We expected that individuals who pay the largest energetic costs—low-ranking adult females and high-ranking adult males (*Anderson et al., 2021*)—would appear old-for-age.

In terms of early-life adversity, prior research in Amboseli has identified six conditions whose cumulative, and sometimes individual, effects predict adult female mortality, including maternal loss prior to age 4 years, drought in the first year of life, birth into an especially large social group, the presence of a close-in-age competing younger sibling, and having a low-ranking or socially isolated mother (*Gesquiere et al., 2011*; *Archie et al., 2012*; *Tung et al., 2016*). For adult female baboons, experiencing multiple sources of adversity in early life is the strongest socio-environmental predictor of mortality; hence, we expected that these individuals would have old-for-age clock estimates in adulthood (*Archie et al., 2014a*; *Archie et al., 2014b*; *Gesquiere et al., 2011*; *Lea et al., 2015*; *Tung et al., 2016*; *Lea et al., 2018*). However, we also expected that some sources of early-life adversity might be linked to young-for-age gut microbiomes in juvenile baboons. For instance, famine is linked to gut microbial immaturity (*Smith et al., 2013*; *Subramanian et al., 2014*; *Blanton et al., 2016*; *Gehrig et al., 2019*), and maternal social isolation might delay gut microbiome development due to less frequent microbial exposures from conspecifics. In contrast to the expectation that harsh early-life conditions are linked to old-for-age microbiomes in adult baboons, a viable alternative is that gut microbiome age might be better predicted by an individual's current environmental or social conditions (e.g., season or social status), rather than past events. Indeed, gut microbiomes are highly dynamic and can change rapidly in response to host diet or other aspects of host physiology, behavior, or environments (*Hicks et al., 2018*; *Kolodny et al., 2019*; *Risely et al., 2021*). Such results would support recency models for biological aging (*Kuh et al., 2003*; *Shanahan et al., 2011*) and would be consistent with findings from a recent epigenetic clock study in Amboseli (*Anderson et al., 2021*).

We began our analyses by identifying gut microbiome features that change reliably with host age. We then constructed a microbiome clock by comparing the performance of several supervised machine learning algorithms to predict host age from gut microbial composition in each sample from each host. We evaluated the clock's performance for male and female baboons and tested whether deviations from clock performance were predicted by the baboons' social and environmental conditions (guided by the predictions outlined above). Lastly, we tested whether baboons with young-for-age gut microbiomes have correspondingly late developmental timelines or longer lifespans. In general, our results support the idea that a baboon's current socio-environmental conditions, especially their current social rank and the season of sampling, have stronger effects on microbiome age than early-life events—many of which occurred many years prior to sampling. As such, the dynamism of the gut microbiome may often overwhelm and erase early-life effects on gut microbiome age. Our work highlights the diversity of ways that social and environmental conditions shape microbiome aging in natural systems.

## Results

### Many microbiome features change with age

Before creating the microbiome clock, we characterized microbiome features that change reliably with the age of individual hosts. Our subjects were 479 known-age baboons (264 females and 215 males) whose microbiome taxonomic compositions were characterized using 13,476 fecal-derived 16S rRNA gene sequencing profiles over a 14-year period (*Figure 1A, B*; baboon age ranged from 7 months to 26.5 years; 8245 samples from females; 5231 samples from males; range = 3–135 samples per baboon; mean = 35 samples per female and 26 samples per male).

We tested age associations for 1440 microbiome features, including: (1) five metrics of alpha diversity; (2) the top 10 principal components (PCs) of Bray–Curtis dissimilarity (which collectively explained 57% of the variation in microbiome community composition); and (3) centered log-ratio (CLR) transformed abundances (*Gloor et al., 2017*) of each microbial phylum ($n = 30$), family ($n = 290$), genus ($n = 747$), and amplicon sequence variance (ASV) detected in >25% of samples ($n = 358$; *Supplementary file 1A*; see methods for details on CLR transformations). We tested these different taxonomic levels in order to learn whether the degree to which coarse and fine-grained designations categories were associated with host age. For each of these 1440 features, we tested its association with host age

by running linear mixed effects models that included linear and quadratic effects of host age and four other fixed effects: sequencing depth, the season of sample collection (wet or dry), the average maximum temperature for the month prior to sample collection, and the total rainfall in the month prior to sample collection (*Grieneisen et al., 2021*; *Björk et al., 2022*; *Tung et al., 2015*). Baboon identity, social group membership, hydrological year of sampling, and sequencing plate (as a batch effect) were modeled as random effects.

We found that many aspects of microbiome community composition changed with host age (*Figure 2*; *Figure 2—figure supplement 1*). All alpha diversity metrics, except richness, the only unweighted metric, exhibited U-shaped relationships with age, with high values in early life and old age, and low values in young adulthood. While we should interpret this pattern with caution due to the small sample size beyond age 20 ($n$ = 18 females), this U-shaped pattern differs somewhat from patterns in humans and chimpanzees: most human populations exhibit an asymptotic increase in alpha diversity with age (*Badal et al., 2020*; *Caporaso et al., 2011*) while in chimpanzees alpha diversity is highest in early life (*Reese et al., 2021*) (false discovery rate [FDR] <0.05; *Figure 2C, E*; *Supplementary file 1A*). Further, seven of the ten PCs of microbiome composition exhibited linear, and in some cases quadratic, relationships with age, with PC1, PC2, PC4, and PC6 exhibiting the strongest age associations (FDR <0.05; *Figure 2C, F*; *Supplementary file 1A*).

51.6% of the 1440 features exhibited significant linear or quadratic relationships with host age (*Figure 2—figure supplement 1* and *Figure 2—figure supplement 2*; *Supplementary file 1A*; FDR <0.05). 60% of phyla (18 of 30) decreased proportionally with age, while only three phyla—Kiritimatiellaeota, Firmicutes, and Chlamydiae—increased proportionally with age (FDR <0.05; *Supplementary file 1A*). Similarly, 66% (66 of 100) of age-associated families and 55% (115 of 209) of age-associated genera exhibited declining proportions with age (*Supplementary file 1A*). Consistent with the idea that age-related taxa differ across host populations and host taxa, none of the taxa that changed in this baboon population were commonly age associated in humans (*Badal et al., 2020*). The taxa most consistently linked to human aging include *Akkermansia*, *Faecalibacterium*, *Bacteroidaceae*, and *Lachnospiraceae* (*Wilmanski et al., 2021*; *Badal et al., 2020*) while in our sample of baboons, the strongest age-related changes were seen in the families *Campylobacteraceae*, *Clostridiaceae*, *Elusimicrobiaceae*, *Enterobacteriaceae*, *Peptostreptococcaceae*, and an uncharacterized family within Gastranaerophilales (*Figure 2D, F*; *Supplementary file 1A*). The genera that had the strongest relationships with age included *Camplylobacter*, *Catenibacterium*, *Elusimicrobium*, *Prevotella*, *Romboutsia*, and *Ruminococcaceae UCG-011* (*Figure 2D, F*; *Supplementary file 1A*).

## Microbiome clock calibration and composition

We next turned our attention to building a microbiome clock using all 9575 microbiome compositional and taxonomic features present in at least 3 samples (*Supplementary file 1B*; the 1440 features discussed above only included taxa present in >25% of samples). We included all 9575 microbiome features in our age predictions, as opposed to just those that were statistically significantly associated with age because removing these non-significant features could exclude features that contribute to age prediction via interactions with other taxa. In developing the clock, we compared the performance of three supervised machine learning methods to predict the chronological age of individual hosts at the time each microbiome sample was collected. The three machine learning methods were elastic net regression, Random Forest regression, and Gaussian process (GP) regression (see Appendix). Because gut microbiomes are highly personalized in ours and other host populations (*Björk et al., 2022*), at least one sample from each host was always present in the training sets for these models (see Methods).

We found that the most accurate age predictions were produced by a GP regression model with a kernel customized to account for heteroscedasticity (*Figure 3*; *Figure 3—figure supplement 1*; *Figure 3—figure supplement 2*). This model predicted host chronological age ($age_c$), with an adjusted $R^2$ of 48.8% and a median error of 1.96 years across all individuals and samples (*Figure 3A*, *Table 1*). As has been observed in some previous age clocks (e.g., *Tian et al., 2017*; *Declerck and Vanden Berghe, 2018*; *de la Cuesta-Zuluaga et al., 2019*), microbial age estimates ($age_m$) were compressed relative to the $x = y$ line, leading the model to systematically over-predict the ages of young individuals and under-predict the ages of old individuals (*Figure 3A*).

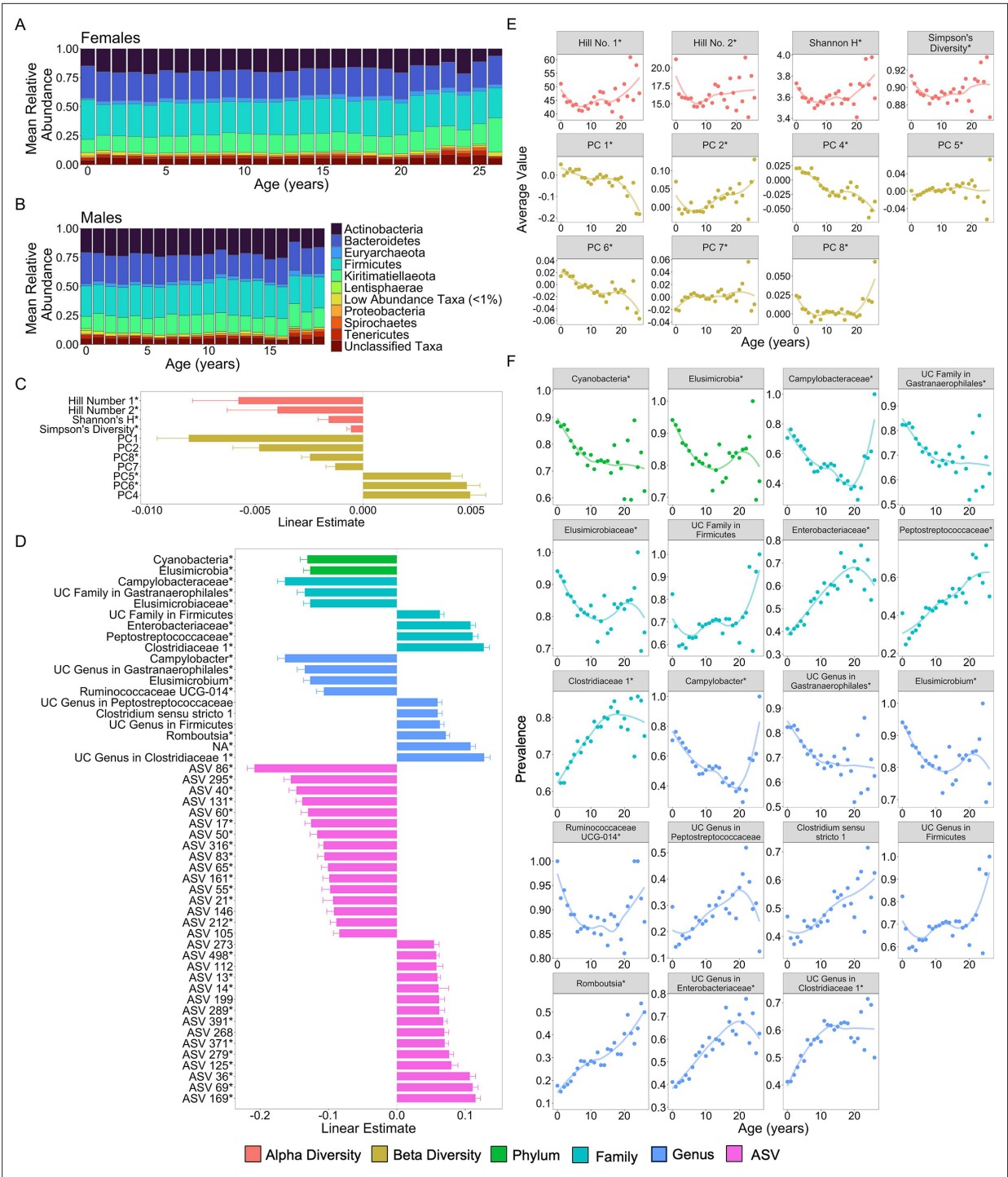

**Figure 2.** Microbiome features change with age. (**A**) and (**B**) show the percent mean relative abundance of microbial phyla across life for females and males, respectively. Panel (**C**) shows the estimates of the linear associations between mean-centered age for metrics of microbiome alpha diversity and principal components of microbiome compositional variation that exhibited significant associations with age (false discovery rate [FDR] <0.05). Positive values are more abundant in older hosts. Panel (**D**) shows the estimate of the linear association between mean-centered age and the top 50 microbiome features that exhibited significant associations with age. Positive values are more abundant in older hosts. Panel (**E**) shows the average value of the microbiome features from (**C**) as a function of age, across all subjects. Note that sample sizes for patterns beyond age 20 years rely on 256 samples from just 18 females; hence, we interpret the pattern in these oldest animals with caution. Panel (**F**) shows the average prevalence of the higher taxonomic designations from (**D**) as a function of age, across all subjects. In (**C–F**), points are colored by the category of the feature (see legend). UC is an

*Figure 2 continued on next page*

*Figure 2 continued*

abbreviation for uncharacterized. Features that had a significant quadratic age term are indicated by * (see also *Figure 2—figure supplements 1 and 2*; *Supplementary file 1A*).

The online version of this article includes the following figure supplement(s) for figure 2:

**Figure supplement 1.** The number and proportion of each type of feature that was significantly associated with age.

**Figure supplement 2.** Taxa with the strongest quadratic associations with age.

When we subset our $age_m$ estimates by sex, we found that the microbiome clock was slightly more accurate for males than for females (*Figure 3B*, *Table 1*). The adjusted $R^2$ for the correlation between $age_c$ and $age_m$ for males was 50.0%, with a median prediction error of 1.71 years as compared to an adjusted $R^2$ of 48.9% and median error of 2.15 years for female baboons (*Table 1*). Male baboons also exhibit significantly older gut microbial age than females (*Figure 3B*, chronological age by sex interaction: $\beta = 0.18$, $p < 0.001$, *Supplementary file 1C*). Across the lifespan, males show a 1.4-fold higher rate of change in $age_m$ as a function of $age_c$ compared to females (relationship between $age_c$ and $age_m$ in males: $\beta = 0.63$, $p < 0.001$; relationship between $age_c$ and $age_m$ in females: $\beta = 0.45$, $p < 0.001$; *Supplementary file 1D*). Similar to patterns from a recent epigenetic age-predicting clock developed for this population (*Anderson et al., 2021*), this effect was only present after sexual maturity: when we subset the $age_m$ estimates to microbiome samples collected prior to the median age at sexual maturity (5.4 years for testicular enlargement in males and 4.5 years for menarche in females; *Altmann et al., 2010*), we found no significant interaction between sex and age ($age_c$ by host sex interaction prior to median age of maturity: $\beta = -0.09$, $p = 0.203$; $age_c$ by host sex interaction after median age

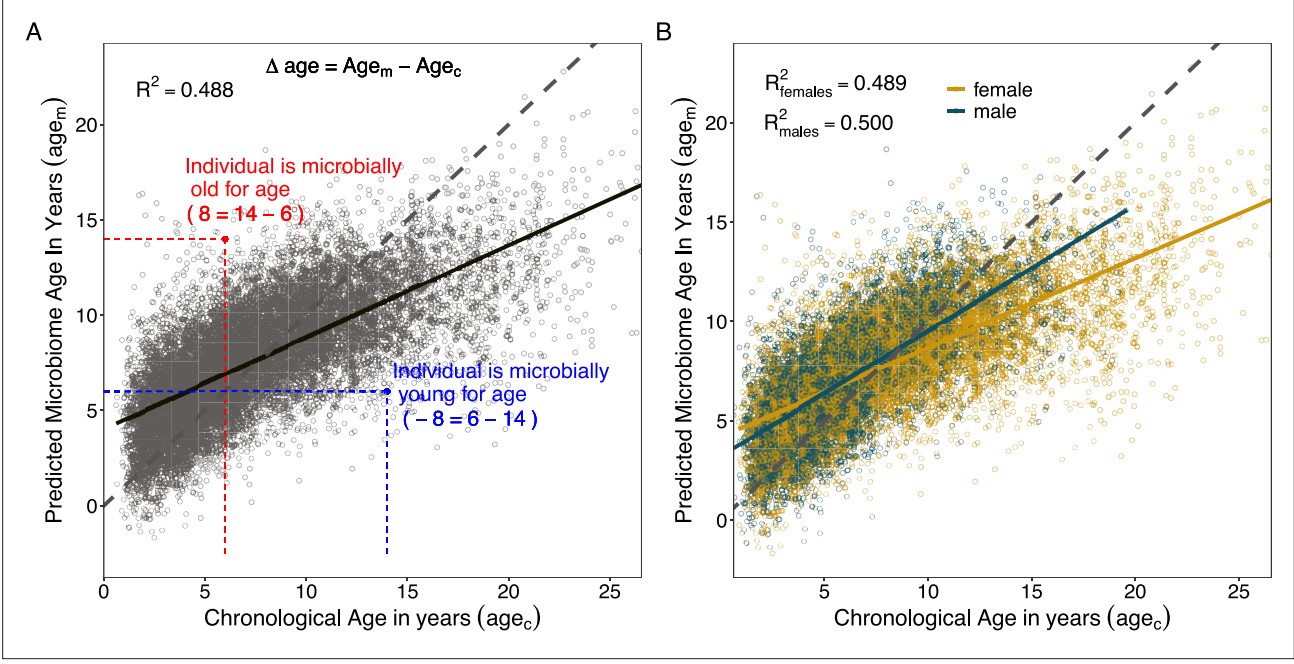

**Figure 3.** Microbiome clock age predictions in wild baboons. Panels (**A**) and (**B**) show predicted microbiome age in years ($age_m$) from a Gaussian process regression model, relative to each baboon's true chronological age in years ($age_c$) at the time of sample collection. Each point represents a microbiome sample. Panel (**A**) shows linear fit for all subjects in the model; (**B**) shows separate linear fits for each sex (*Supplementary file 1D*). Dashed lines show the 1-to-1 relationship between $age_c$ and $age_m$. Panel (**A**) also shows the measurement of sample-specific microbiome Δage compared to chronological age. Whether an estimate is old- or young-for-chronological age is calculated for each microbiome sample as the difference between $age_m$ and $age_c$. Because of model compression relative to the 1-to-1 line, we correct for host chronological age by including chronological age in any model. An example of an old-for-age sample is shown as a red point, with dashed lines showing the value of $age_c$ for a given sample with its corresponding $age_m$.

The online version of this article includes the following figure supplement(s) for figure 3:

**Figure supplement 1.** Microbiome clocks from an ensemble of machine learning algorithms.

**Figure supplement 2.** Variance in residuals across lifespans in the Gaussian process regression prior to correction.

**Table 1.** Comparison of Gaussian process regression model performance between sexes. Model accuracy was determined based on the correlation between known chronological age (age$_c$) and predicted age (age$_m$), the variance explained in age$_c$ by age$_m$ ($R^2$), and the median absolute difference between age$_c$ and age$_m$ (*Horvath, 2013*).

| Subset | Sample size | $R^2$ | Pearson's R | Median error (years) |
|---|---|---|---|---|
| All subjects | 13,476 | 48.8% | 0.698 | 1.962 |
| Females only | 8245 | 48.9% | 0.699 | 2.150 |
| Males only | 5231 | 50.0% | 0.707 | 1.706 |

of maturity: $\beta$ = 0.15, p < 0.001; *Supplementary file 1C*). After maturity, we recapitulate the 1.4-fold higher rate of change in males compared to females observed in the full dataset (relationship between age$_c$ and age$_m$ in males only: $\beta$ = 0.53, p < 0.001; relationship between age$_c$ and age$_m$ in females only: $\beta$ = 0.38, p < 0.001; *Supplementary file 1D*).

Overall, age$_m$ estimates performed reasonably well compared to other known predictors of age in the Amboseli baboons (*Supplementary file 1E*; *Declerck and Vanden Berghe, 2018*). Age$_m$ performed similarly to the non-invasive physiology and behavior clock (NPB clock) recently developed for this population (*Weibel et al., 2024*; $R^2$ = 51%; median error = 2.33 years). Age$_m$ was a stronger predictor of host chronological age than body mass index (BMI; except juvenile male BMI), blood cell composition from flow cytometry, and differential white blood cell counts from blood smears (*Supplementary file 1E*; *Anderson et al., 2021*; *Altmann et al., 2010*, *Weibel et al., 2024*; *Jayashankar et al., 2003*; *Galbany et al., 2011*). However, age$_m$ was a less accurate predictor of chronological age than both dentine exposure (males and females, respectively: adjusted $R^2$ = 73%, 85%; median error = 1.11 and 1.12 years; *Supplementary file 1E*) and an epigenetic clock based on DNA methylation data (males and females, respectively: adjusted $R^2$ = 74%, 60%; median error = 0.85 and 1.62 years; *Supplementary file 1E*; and *Declerck and Vanden Berghe, 2018*).

## Social and environmental conditions predict variation in microbiome age

To test whether deviations in microbiome age for a given chronological age are correlated with socio-environmental predictors of health and mortality risk, we calculated whether microbiome age estimates from individual samples were older or younger than their hosts' known chronological ages (Δage in years; *Figure 3A*). We then tested whether several social and environmental variables predicted individual variation in microbiome Δage in years (*Supplementary file 1G*; note that whether microbiome ages are old- or young-for-age is correlated with host age, hence our models always included host chronological age as a covariate). Overall, we expected that adult baboons who experienced harsh conditions in early-life adversity (the strongest socio-environmental predictor of adult mortality in Amboseli) would tend to look old-for-age based on the microbiome clock (*Archie et al., 2014a*; *Archie et al., 2014b*; *Gesquiere et al., 2011*; *Lea et al., 2015*; *Tung et al., 2016*; *Lea et al., 2018*). Alternatively, microbiome deviations from chronological age might be best predicted by an individual's current social status or season, rather than past events. These results would support recency models of biological aging (*Kuh et al., 2003*; *Shanahan et al., 2011*) and would be consistent with findings from a recent epigenetic clock study in Amboseli (*Anderson et al., 2021*).

We found that individual baboons varied considerably in gut microbiome Δage. For instance, in mixed effects models, individual identity explained ~25% to ~50% of the variance in Δage for females and males, respectively, over the course of their lives (*Supplementary file 1F*). Further, we found that season, dominance rank, and some aspects of early-life adversity (large group size, drought, and maternal social isolation) were linked to small deviations from chronological age. In support of our expectation that microbiome samples collected in the dry season are old-for-chronological age, we found that age estimates based on microbiome samples collected from female baboons in the dry season were ~2 months older than the host's true chronological age ($\beta$ = −0.180, p = 0.021, *Table 2*; *Supplementary file 1C–F*; *Figure 4—figure supplement 1A*). However, season did not significantly predict the difference between microbiome age and known age in male baboons.

In terms of social status, we expected to observe that low-ranking females and high-ranking males would be old-for-chronological age (*Anderson et al., 2021*). In support, we found that estimates

**Table 2.** Social and environmental factors predicting variation in microbiome Δage in female and male baboons.

Models below only show variables that minimize the Akaike information criterion (AIC) for each model; see *Supplementary file 1F* for full models. Coefficients for social dominance rank are transformed so higher values reflect higher rank/social status (see footnotes).

| Fixed effect | β | p-value | Interpretation |
|---|---|---|---|
| *Predictors of microbiome Δage in females (n = 6743 samples from 192 females)* | | | |
| Chronological age | –0.551 | <0.001 | Included to control for the correlation between chronological age and microbiome Δage (*Figure 3*) |
| Season | –0.180 | 0.021 | Dry season samples are microbially old-for-age |
| Proportional rank* | 1.745 | <0.001 | Low-ranking females are microbially young-for-age |
| *Predictors of microbiome Δage in males (n = 4355 samples from 168 males)* | | | |
| Chronological age | –0.404 | <0.001 | Included to control for the correlation between chronological age and microbiome Δage (*Figure 3*) |
| Ordinal rank† | 0.033 | <0.001 | Low-ranking males are microbially young-for-age |
| Born in a drought | –0.451 | 0.021 | Males born during a drought are microbially young-for-age |
| Born into a large group | 0.471 | 0.033 | Males born into large groups are microbially old-for-age |
| Socially isolated mother | –0.395 | 0.006 | Males with a socially isolated mother are microbially young-for-age |

*Proportional rank ranges from 0 to 1, with higher values reflecting higher social status.

†Ordinal rank is an integer ranking, with lower values reflecting higher social status; we have inverted the sign of the coefficient so higher numbers reflect higher rank to facilitate comparison to females.

from high-ranking males were old-for-age compared to estimates from low-ranking males, but these effects were relatively weak and noisy (rank effect: $\beta$ = 0.033, p < 0.001; *Figure 4A*; *Table 2*; *Supplementary file 1C–F*). Specifically, controlling for chronological age, alpha male gut microbiomes (ordinal rank = 1) appeared to be approximately 4 months older than microbiomes sampled from males with an ordinal rank of 10 (*Table 2*). However, contrary to our expectations, high-ranking female baboons also had old-for-age estimated when compared to low-ranking females (rank effect: $\beta$ = 1.745, p < 0.001; *Figure 4B*; *Table 2*; *Supplementary file 1A–F*). Specifically, controlling for chronological age, the microbiome of an alpha female (proportional rank = 1) appeared to be approximately 1.75 years older than the lowest ranking females in the population (proportional rank = 0; *Table 2*).

Some forms of early-life adversity also predicted variation in microbiome Δage, but only in males, and in inconsistent directions. For instance, males born into the highest quartile of observed group sizes had old-for-age estimates. Males experiencing this source of early-life adversity had gut microbiomes that were predicted to be ~5.4 months older than males not experiencing this source of adversity ($\beta$ = 0.471, p = 0.033, *Table 2*; *Supplementary file 1D–F*; *Figure 4—figure supplement 1C*). However, early-life drought and maternal social isolation were linked to young-for-age gut microbiomes in males (drought effect: $\beta$ = –0.451, p = 0.021; maternal social isolation effect: $\beta$ = –0.395, p = 0.006, *Table 2*; *Supplementary file 1D–F*; *Figure 4—figure supplement 1D*). Probably as a result of these conflicting effects, we found no effect of cumulative early adversity on microbiome Δage in males (*Supplementary file 1B–F*).

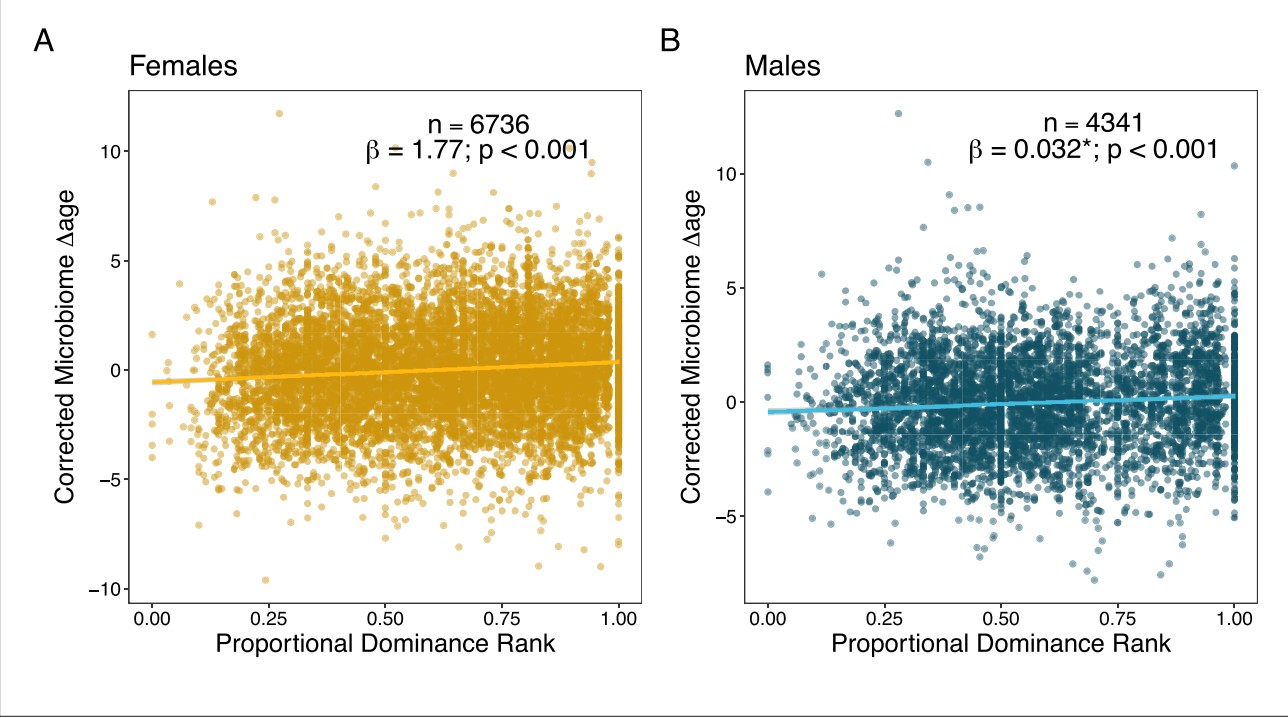

**Figure 4.** Social dominance rank predicts gut microbiome Δage in male and female baboons (corrected for confounders). Panels (**A**) and (**B**) show the relationship between host proportional dominance rank and corrected gut microbiome Δage in (**A**) males (blue points) and (**B**) females (yellow points). Each point represents an individual gut microbiome sample. Corrected microbiome Δage is calculated as the residuals of $age_m$ correcting for host chronological age, season, monthly temperature, monthly rainfall, and social group and hydrological year at the time of collection.

The online version of this article includes the following figure supplement(s) for figure 4:

**Figure supplement 1.** Statistically significant socio-environmental predictors of corrected Δage not shown in *Figure 4* in the main text.

## Microbiome age does not predict the timing of development or survival

Finally, we tested whether variation in microbiome Δage predicted the timing of individual maturational milestones or survival using Cox proportional hazards models (*Supplementary file 1H*). Maturational milestones for females were the age at which they attained their first adult dominance rank, reached menarche, or gave birth to their first live offspring (*Figure 1C*). Male maturational milestones were the age at which they attained testicular enlargement, dispersed from their natal social group, or attained their first adult dominance rank (*Figure 1D*). We also tested if microbiome Δage predicted juvenile survival (in females and males) or adult survival (females only). We did not test adult survival in males because male dispersal makes it difficult to know age at death for most males (*Campos et al., 2020*).

Contrary to our expectations, microbiome Δage did not predict the timing of any baboon developmental milestone or measure of survival (*Supplementary file 1I, J*). However, these patterns should be treated with caution, as reflected by the large number of censored animals, large hazard ratios, and small sample sizes for some tests.

## Discussion

We report three main findings. First, similar to humans and other animals (*Bäckhed et al., 2005*; *Mueller et al., 2006*; *Koenig et al., 2011*; *Yatsunenko et al., 2012*; *Bergström et al., 2014*; *Langille et al., 2014*; *Clark et al., 2015*; *Cong et al., 2016*; *Yassour et al., 2016*; *Biagi et al., 2016*; *Odamaki et al., 2016*; *Smith et al., 2017*; *Reese et al., 2021*; *Baniel et al., 2021*), baboon gut microbiomes show age-related changes in taxonomic composition that produce a dependable microbiome-based age predictor—a microbiome clock. This clock explains nearly half the variance in true host

chronological age, and variation in its age predictions recapitulate well-known patterns of faster male senescence in humans and other primates (*Lemaître et al., 2020*). Second, parallel to a recent epigenetic clock in the Amboseli baboons (*Anderson et al., 2021*), deviations from microbiome age predictions are explained by socio-environmental conditions experienced by individual hosts, especially recent conditions, although the effect sizes are small and are not always directionally consistent. Notably, recent social competition as reflected in social dominance rank predicts both microbiome and epigenetic age. Third, microbiome age did not predict the timing of individual development or survival (but caution is warranted because sample sizes were small for some tests). Hence, in our host population, gut microbial age reflects current social and environmental conditions, but not necessarily the pace of development or mortality risk.

To date, at least five other microbiome clocks have been built—all in human subjects—that predict host chronological age (*de la Cuesta-Zuluaga et al., 2019*; *Galkin et al., 2020*; *Huang et al., 2020*; *Chen et al., 2022*). Compared to these clocks, our clock in baboons has comparable or better predictive power, with a median error of 1.96 years, compared to 6–11 years in human age-predicting clocks (*de la Cuesta-Zuluaga et al., 2019*; *Galkin et al., 2020*; *Huang et al., 2020*; *Chen et al., 2022*; baboon lifespans are approximately one third of a human lifespan). Age prediction from gut microbial composition may be more successful in baboons than humans for at least three reasons. First, signs of age in the baboon gut microbiome may be more consistent across hosts, perhaps because of the relative homogeneity in host environments and lifestyles in baboons compared to humans (*Björk et al., 2022*). Second, the baboon clock relies on dense longitudinal sampling for each host, and because the training dataset included at least one sample from each host, the microbiome clock may be better able to address personalized microbiome compositions and dynamics than clocks that rely on cross-sectional data (e.g., *de la Cuesta-Zuluaga et al., 2019*; *Galkin et al., 2020*; *Huang et al., 2020*). However, because our training dataset was not naive to information from the host being predicted, this approach could leak information between the training and test set. Third, our GP regression approach allowed us to account for non-linear and interactive relationships between microbes with age, leveraging a wider variety of age-related signatures in the microbiome than other machine learning approaches (e.g., elastic net regression or Random Forest regression).

Despite the relative accuracy of the baboon microbiome clock compared to similar clocks in humans, our clock has several limitations. First, the clock's ability to predict individual age is lower than for age clocks based on patterns of DNA methylation—both for humans and baboons (*Horvath, 2013*; *Marioni et al., 2015*; *Chen et al., 2016*; *Binder et al., 2018*; *Anderson et al., 2021*). One reason for this difference may be that gut microbiomes can be influenced by several non-age-related factors, including social group membership, seasonal changes in resource use, and fluctuations in microbial communities in the environment. Second, gut microbiomes are highly personalized: each host species, population, and even host individuals within populations have distinctive, characteristic microbiomes, which likely limits the utility of our clock beyond our study population (*Björk et al., 2022*; *Grieneisen et al., 2017*; *Tung et al., 2015*; *Grieneisen et al., 2019*). To make a more generalizable clock, an important next step would be to train the clock on data from many more host populations and incorporate features of the gut microbiome that are broader and more universal across host species and populations. Third, the relationships between potential socio-environmental drivers of biological aging and the resulting biological age predictions were inconsistent. For instance, some sources of early-life adversity were linked to old-for-age gut microbiomes (e.g., males born into large social groups), while others were linked to young-for-age microbiomes (e.g., males who experienced maternal social isolation or early-life drought), or were unrelated to gut microbiome age (e.g., males who experienced maternal loss; any source of early-life adversity in females).

The most consistent socio-ecological driver of baboon microbiome age was individual dominance rank. Microbiome samples from high-ranking males and females both appeared old-for-age. These results are interesting considering a growing body of evidence that finds rank-related differences in immunity and metabolism, including costs of high rank, especially for males (*Anderson et al., 2021*; *Gesquiere et al., 2011*; *Archie et al., 2012*; *Anderson et al., 2022*; *Habig and Archie, 2015*; *Snyder-Mackler et al., 2016*). For instance, in Amboseli, high social status in males is linked to old-for-age epigenetic age estimates (*Anderson et al., 2021*), differences in immune regulation (*Archie et al., 2012*; *Anderson et al., 2022*), and, for alpha males, elevated glucocorticoid levels (*Gesquiere et al., 2011*). These patterns, together with our evidence that high-ranking males tend to look 'old-for-age',

are consistent with the idea that high-ranking males pursue a 'live fast die young' life history strategy (*Anderson et al., 2021*).

However, we also found evidence for old-for-age microbiome age estimates in samples from high-ranking females who do not seem to be 'living fast' in the same sense as high-ranking males (indeed alpha females have *lower* glucocorticoid hormones than other females; *Levy et al., 2020a*). This outcome points toward a shared driver of high social status in shaping gut microbiome age in both males and females. While it is difficult to identify a plausible shared driver, one benefit shared by both high-ranking males and females is priority of access to food. This access may result in fewer foraging disruptions and a higher quality, more stable diet. At the same time, prior research in Amboseli suggests that as animals age, their diets become more canalized and less variable (*Grieneisen et al., 2021*). Hence aging and priority of access to food might both be associated with dietary stability and old-for-age microbiomes. However, this explanation is speculative and more work is needed to understand the relationship between rank and microbiome age.

While some social and environmental conditions associated with baboon development or mortality predict microbiome age, our microbiome clock predictions do not themselves predict baboon development or mortality. This finding supports the idea that microbiome age is sensitive to transient social and environmental conditions; however, these patterns do not have long-term consequences for development and mortality. One reason for this may be that the biological drivers of development and mortality are too diverse to be well reflected in gut microbial communities. For instance, animals in Amboseli die for many reasons, including interactions with predators and humans, conflict with conspecifics (*Paietta et al., 2022*), and disease, and the biological predictors of these events in the gut microbiome are likely weaker or more diverse than the biological signals that predict developmental milestones (i.e., sex steroids, growth hormones, metabolic status, and physical condition).

Three important future directions of our work will be to test: (1) whether microbiome age is correlated with other hallmarks of biological age in this population; (2) whether it is possible to build a microbiome-based predictor of individual lifespan (i.e., *remaining* lifespan as opposed to years already lived); and (3) the relationships between microbiome compositional features and individual survival. While several metrics of biological age have been developed for this population (*Anderson et al., 2021*; *Altmann et al., 2010*; *Weibel et al., 2024*; *Jayashankar et al., 2003*; *Galbany et al., 2011*), testing the correlations between them is complex because these metrics were measured for different sets of subjects at different ages and with different sampling designs (e.g., longitudinal versus cross-sectional measures). We also hope to measure epigenetic age in fecal samples, leveraging methods developed in *Hanski et al., 2024*.

In sum, our findings support the hypothesis that the gut microbiome is a biomarker of some aspects of host age. By leveraging microbial, social, environmental, and life history data on host individuals followed from birth to death, we bolster the validity of microbiome clock studies in humans and find some socio-environmental predictors of microbiome age. Future work may benefit from searching for more universal aspects of the microbiome that may predict host aging across populations and even host species.

## Materials and methods
### Study population and subjects

Our study subjects were 479 wild baboons (215 males and 264 females) living in the Amboseli ecosystem in Kenya between April 2000 and September 2013. The Amboseli baboon population is primarily composed of yellow baboons (*Papio cynocephalus*) with some admixture from nearby anubis baboon (*Papio anubis*) populations (*Samuels and Altmann, 1986*; *Tung et al., 2008*; *Vilgalys et al., 2022*). Prior research in our population finds no link between host hybrid ancestry and microbiome composition (*Grieneisen et al., 2019*). Since 1971, the Amboseli Baboon Research Project (ABRP) has been collecting continuous observations of the baboons' demography, behavior, and environment (*Alberts and Altmann, 2012*). The baboons are individually identified by expert observers who visit and collect data on each social group 3–4 times per week (the subjects lived in up to 12 different social groups over the study period). During each monitoring visit, the observers conduct group censuses and record all demographic events, including births, maturation events, and deaths, allowing us to calculate age at maturity and lifespan with precision. This research was approved by the IACUC at

Duke University, University of Notre Dame, and Princeton University and the Ethics Council of the Max Planck Society and adhered to all the laws and guidelines of Kenya.

## Sample collection, DNA extraction, and 16S data generation

The 13,476 gut microbiome compositional profiles in this analysis represent a subset of 17,277 profiles, which were previously described in *Grieneisen et al., 2021*; *Björk et al., 2022*. The 13,476 samples in our analyses include those from baboons whose birthdates, and hence individual ages, were known with just a few days' error. Each baboon had on average 33 samples collected across 6 years of their life (*Figure 1A, B*; range = 3–135 samples per baboon; median days between samples = 44 days).

Samples were collected from the ground within 15 min of defecation. For each sample, approximately 20 g of feces was collected into a paper cup, homogenized by stirring with a wooden tongue depressor, and a 5-g aliquot of the homogenized sample was transferred to a tube containing 95% ethanol. While a small amount of soil was typically present on the outside of the fecal sample, mammalian feces contains 1000 times the number of microbial cells in a typical soil sample (*Sender et al., 2016*; *Raynaud and Nunan, 2014*), which overwhelms the signal of soil bacteria in our analyses (*Grieneisen et al., 2019*). Samples were transported from the field in Amboseli to a lab in Nairobi, freeze-dried, and then sifted to remove plant matter prior to long term storage at –80°C.

DNA from 0.05 g of fecal powder was manually extracted using the MoBio (Catalog No. 12955-12) and Qiagen (Catalog No. 12955-4) PowerSoil HTP kits for 96-well plates using a modified version of the MoBio PowerSoil-HTP kit. Specifically, we followed the manufacturers' instructions but increased the amount of PowerBead solution to 950 µl/well and incubated the plates at 60°C for 10 min after the addition of PowerBead solution and lysis buffer C1. We included one extraction blank per batch, which had significantly lower DNA concentrations than sample wells (*t*-test; $t = -50$, p < $2.2 \times 10^{-16}$; *Grieneisen et al., 2021*). We also included technical replicates, which were the same fecal sample sequenced across multiple extraction and library preparation batches. Technical replicates from different batches clustered with each other rather than with their batch, indicating that true biological differences between samples are larger than batch effects.

Following DNA extraction, a ~390-bp segment of the V4 region of the 16S rRNA gene was amplified and libraries prepared following standard protocols from the Earth Microbiome Project (*Gilbert et al., 2014*). Libraries were sequenced on the Illumina HiSeq 2500 using the Rapid Run mode (2 lanes per run). Sequences were single indexed on the forward primer and 12 bp Golay barcoded. The resulting sequencing reads were processed following a DADA2 pipeline (*Callahan et al., 2016*), with the following additional quality filters: we removed samples with low DNA extraction concentrations (<4× the plate's blank DNA extraction concentration), samples with <1000 reads, and amplicon sequence variants that appeared in one sample (see (*Grieneisen et al., 2021*) for details). ASVs were assigned to microbial taxa using the `IdTaxa(…)` function in the DECIPHER package, against the Silva reference database SILVA_SSU_r132_March2018.RData (*Wright et al., 2012*; *Quast et al., 2013*). The final set of samples had 1017–427,454 reads (median = 51,839 reads), with 8492 total ASVs.

## Identifying microbiome features that contribute to age predictions and that change with age

To identify microbiome features that change with host age, we ran linear mixed models on 1440 microbiome features (*Supplementary file 1A*). Models were run using the R package *lme4*, with p-value estimates from *lmerTest* (*Bates et al., 2015*; *Kuznetsova et al., 2017*). These features included: (1) five metrics of alpha diversity; (2) the top 10 PCs of microbiome compositional variation; (3) CLR transformed abundances (i.e., read counts) of each microbial phyla (*n* = 30), family (*n* = 290), genus (*n* = 747), and ASV detected in >25% of samples (*n* = 358). CLR transformations are a recommended approach for addressing the compositional nature of 16S rRNA amplicon read count data (*Gloor et al., 2017*). Alpha diversity metrics were calculated using the R package *vegan* and PCs of microbiome compositional variation were calculated using the R package *labdsv* (*Dixon, 2003*; *Roberts, 2019*).

For each feature, we modeled its relationship to host chronological age using both linear and quadratic terms. To make our quadratic terms more interpretable, we centered our age estimates on zero by subtracting the average age in the dataset from each age value. Specifically, when a quadratic term is negative, the curve is concave, whereas when the term is positive, the curve is convex. We also

included season (wet or dry) and *z*-scored read count, rainfall, and temperature as fixed effects, and individual identity, social group at time of collection, hydrological year, and the DNA extraction/PCR plate identity were modeled as random effects. We did not model individual social network position because prior analyses of this dataset find no evidence that close social partners have more similar gut microbiomes, probably because we lack samples from close social partners sampled close in time (*Grieneisen et al., 2021*; *Björk et al., 2022*). All community features (i.e., alpha diversity and PCs) and all taxa present in >25% of samples were modeled using a Gaussian error distribution. We extracted the coefficient, standard error, and p-value for the age term, then corrected for multiple testing using the FDR approach of *Benjamini and Hochberg, 1995*.

## Building the gut microbiome clock

We created a microbiome clock by fitting a GP regression model (with a kernel customized to account for heteroskedasticity) to predict each baboon's chronological age at the time of sample collection using 9575 microbiome compositional and taxonomic features present in at least three samples (*Supplementary file 1B*; i.e., we did not restrict the features in the clock to the 1440 most abundant features used in the age-association analyses described above). The GP regression model with heteroskedasticity correction was the best performing of four supervised machine learning approaches we considered, including elastic net, Random Forest, and GP regression with and without the heteroskedasticity kernel (*Figure 3—figure supplement 1*; *Supplementary file 1K*; see Appendix for a comparison of other algorithms). Pearson's correlations between age predictions across the four methods ranged from 0.69 between the Random Forests and the GP regression model without the heteroskedasticity kernel to 0.96 between the two GP regression models (with and without the heteroskedasticity kernel; *Supplementary file 1K*).

GP regressions were conducted in Python 3 using scikit-learn (*Van Rossum and Drake, 2009*; *Pedregosa et al., 2011*). As a nonparametric, Bayesian approach that infers a probability distribution over all the potential functions that fit the data, the GP regression does not assume a linear relationship between chronological and predicted age (*Rasmussen and Williams, 2005*). For the prior distribution in the GP regression, we used a radial basis function as our kernel and set the scale parameter to the mean Euclidean distance of the dataset, as calculated in the R package vegan (*Dixon, 2003*). Because initial, exploratory models exhibited heteroskedasticity (*Figure 3—figure supplement 2*), we multiplied the variance in the training data by the radial basis function, which distributed the higher variance in later life more evenly across lifespan.

To calculate a microbial age estimate for every sample, and to estimate generalization error, we used nested fivefold cross-validation. In each of the five model runs, we used 80% of the data to train the model, and the remaining 20% of the dataset as the test data. Because host identity has a strong effect on microbiome composition in our population (*Björk et al., 2022*), we distributed samples from each host across the five test/training datasets by randomly assigning each sample a test set without replacement. Hence, the training dataset was not naive to information from the given host being predicted as some samples for that host were included in the training set. For each model run, four of the test datasets were treated altogether as training data and the fifth set was the validation test set. We then took the estimates from all five model runs and estimated global model accuracy on the aggregated estimates.

We assessed the accuracy of our microbiome clock by regressing each sample's chronological age (age$_c$) against the model's predicted microbial age (age$_m$) and determining the $R^2$ value and Pearson's correlation between age$_c$ and age$_m$. We also calculated the median error of the model fit as the median absolute difference between age$_c$ and age$_m$ across all samples (*Horvath, 2013*).

## Calculating microbiome Δage estimates

To characterize patterns of microbiome age from our microbiome clock, we calculated sample-specific microbiome Δage in years as the difference between a sample's microbial age estimate, age$_m$ from the microbiome clock, and the host's chronological age in years at the time of sample collection, age$_c$. Higher microbiome Δages indicate old-for-age microbiomes, as age$_m$ > age$_c$, and lower values (which are often negative) indicate a young-for-age microbiome, where age$_c$ > age$_m$ (see *Figure 3*). Because the microbiome clock systematically over predicted the ages of young animals and under predicted the ages of old animals, we also calculated a 'corrected microbiome Δage' as the residuals of age$_m$

correcting for host chronological age, season, monthly temperature, monthly rainfall, and social group and hydrological year at the time of collection. This measure is used for visualizations of the predictors of microbiome age and for testing whether average microbiome Δage predicts developmental milestones or survival.

## Testing sources of variation in microbiome Δage

Many social and environmental factors have been shown to predict fertility and survival in the Amboseli baboons (*Altmann et al., 2010*; *Archie et al., 2014b*; *Tung et al., 2016*; *Gesquiere et al., 2018*; *Altmann and Alberts, 2005*). To test if some of the most important known factors also predict patterns of microbiome age, we used linear mixed models to test predictors of microbiome age in individual samples separately for males and females.

In these models, the response variable was the sample-specific measure of Δage ($age_m$ – $age_c$). All models included the following fixed effects: individual chronological age at the time of sample collection, to correct for model compression; the average maximum temperature during the 30 days before the sample was collected, total rainfall during the 30 days before the sample was collected, and the season (wet or dry) during sample collection (*Supplementary file 1L*). Every model also included, as fixed effects, measures of early-life adversity the individual experienced prior to 4 years of age (*Supplementary file 1G*). These were modeled as either six, individual, binary variables, reflecting the presence or absence of each source of adversity in the first 4 years of life, or as a cumulative sum of the number of sources of adversity the individual experienced, also in the first 4 years of life (*Tung et al., 2016*). Social rank at the time of sampling was also modeled as a fixed effect. For males we used ordinal rank, and for females we used proportional rank (*Levy et al., 2020b*). To make model interpretation more intuitive (high rank corresponds to higher values), we multiplied the coefficients for ordinal rank and maternal rank by –1. Random effects included individual identity, the social group the individual lived in at the time of collection, and hydrological year. In models of microbiome age in females, the number of adult females in the group at the time of sample collection was included as female-specific measure of resource competition.

## Testing whether microbiome Δage predicts baboon maturation and survival

We used Cox proportional hazards models to test whether microbiome Δage predicted the age at which females and males attained maturational milestones and the age at death for juveniles and adult females (*Supplementary file 1H*). We only measured adult survival in females because males disperse between social groups, often repeatedly across adulthood, making it is difficult to know if male disappearances are due to dispersal or death (*Campos et al., 2020*). For females, the maturational milestones of interest were the age at adult rank attainment (median age 2.24 in Amboseli), age at menarche (median age 4.51 in Amboseli), and the age at first live birth (median age 5.97 in Amboseli). For males, these milestones were the age of testicular enlargement (median age 5.38 in Amboseli), the age of dispersal from natal group (median age 7.47 in Amboseli), and the age of first adult rank attainment (i.e., when a male first outranks another adult male in his group's dominance hierarchy; median age 7.38 in Amboseli) (*Charpentier et al., 2008*; *Onyango et al., 2013*). See full descriptions of each milestone in *Supplementary file 1H*. To be included in these analyses, animals must have reached the milestone after the onset of sampling (April 2000) and had at least three samples available in the timeframe of interest. We verified that none of our models violated the proportional hazards assumption of a Cox regression.

The variables we modeled differed based on the event of interest. However, all models included as fixed effects corrected Δage as the residuals of gut microbiome Δage averaged over the timeframe. All models of developmental milestones also included variables tested in *Charpentier et al., 2008* and *Onyango et al., 2013*: (1) maternal presence at the time of the milestone, (2) the number of maternal sisters in the social group, averaged over the timeframe, (3) rainfall averaged over the timeframe, and (4) whether the subject's mother was low ranked (was in the lowest quartile for female ordinal rank). For female-specific milestones, we also included (5) the average number of adult females in the group averaged over the timeframe, and for male-specific milestones we included the number of excess cycling females in the group averaged over the timeframe, or the difference between the number of cycling females and the number of mature males within a subject's social group. Last, we included (6)

the subject's hybrid score, which is an estimation of the proportion of an individual's genetic ancestry attributable to anubis or yellow baboon ancestry (*Vilgalys et al., 2022*).

All juvenile survival models included as fixed effects the residuals of microbiome Δage averaged over the timeframe and measures the cumulative number of sources of early-life adversity each individual experienced (*Tung et al., 2016*). Additionally, we ran three versions of the juvenile survival analysis: two subset to each sex, and one version that included both sexes. In the model including both sexes, we included sex as a predictor.

Adult female survival models included the same variables as for juvenile survival, but additionally included average lifetime dyadic social connectedness to adult females, average lifetime dyadic social connectedness to adult males, and average lifetime proportional rank. Full descriptions of all predictors are available in *Supplementary file 1L*.

## Acknowledgements

We thank Jeanne Altmann for her essential role in stewarding the Amboseli Baboon Project, and in collecting many of the fecal samples used in this manuscript. In Kenya, we thank the Kenya Wildlife Service, the Wildlife Training and Research Institute, the National Council for Science, Technology, and Innovation, and the National Environment Management Authority for permission to conduct research and collect biological samples. We also thank the University of Nairobi, Institute of Primate Research, National Museums of Kenya, the Amboseli-Longido pastoralist communities, the Enduimet Wildlife Management Area, Ker & Downey Safaris, Air Kenya, and Safarilink for their cooperation and assistance in the field. We thank Karl Pinc for managing and designing the database. We thank Raphael Mututua, Serah Sayialel, Kinyua Warutere, and Long'ida Siodi for collecting the field data and fecal samples. We also thank Tawni Voyles, Anne Dumaine, Yingying Zhang, Meghana Rao, Tauras Vilgalys, Amanda Lea, Noah Snyder-Mackler, Paul Durst, Jay Zussman, Garrett Chavez, and Reena Debray for contributing to fecal sample processing. Complete acknowledgments for the ABRP can be found online at https://amboselibaboons.nd.edu/acknowledgements/. This work was supported by the National Institutes of Health and National Science Foundation, especially the National Institute on Aging for R01 AG071684 (EAA), R21 AG055777 (EAA, RB), NIH R01 AG053330 (EAA), NIH R35 GM128716 (RB), NSF DBI 2109624 (MRD), and NSF DEB 1840223 (EAA, JAG). We also thank the Duke University Population Research Institute P2C-HD065563 (pilot to JT), the University of Notre Dame's Eck Institute for Global Health (EAA), and the Notre Dame Environmental Change Initiative (EAA). We also thank Duke University, Princeton University, the University of Notre Dame, the Chicago Zoological Society, the Max Planck Institute for Demographic Research, the L.S.B. Leakey Foundation, and the National Geographic Society for support at various times over the years.

## Additional information

### Competing interests

Jenny Tung: Reviewing editor, eLife. The other authors declare that no competing interests exist.

### Funding

| Funder | Grant reference number | Author |
|---|---|---|
| National Institutes of Health | AG071684 | Elizabeth A Archie |
| National Institutes of Health | AG055777 | Ran Blekhman Elizabeth A Archie |
| National Institutes of Health | AG053330 | Elizabeth A Archie |
| National Institutes of Health | GM128716 | Ran Blekhman |
| National Science Foundation | 2109624 | Mauna R Dasari |

| Funder | Grant reference number | Author |
|---|---|---|
| National Science Foundation | 1840223 | Jack A Gilbert Elizabeth A Archie |
| Duke University | P2C-HD065563 | Jenny Tung |
| University of Notre Dame | Eck Institute for Global Health | Elizabeth A Archie |

The funders had no role in study design, data collection, and interpretation, or the decision to submit the work for publication.

### Author contributions
Mauna R Dasari, Conceptualization, Resources, Data curation, Software, Formal analysis, Funding acquisition, Validation, Investigation, Visualization, Methodology, Writing – original draft, Project administration, Writing – review and editing; Kimberly E Roche, Validation, Methodology, Writing – review and editing; David Jansen, Resources, Data curation, Software, Validation, Visualization, Methodology, Writing – review and editing; Jordan Anderson, Resources, Software, Methodology, Writing – review and editing; Susan C Alberts, Jenny Tung, Resources, Supervision, Funding acquisition, Project administration, Writing – review and editing; Jack A Gilbert, Resources, Funding acquisition, Writing – review and editing; Ran Blekhman, Resources, Funding acquisition, Project administration, Writing – review and editing; Sayan Mukherjee, Resources, Supervision, Methodology, Writing – review and editing; Elizabeth A Archie, Conceptualization, Resources, Data curation, Software, Formal analysis, Supervision, Funding acquisition, Validation, Investigation, Visualization, Methodology, Writing – original draft, Project administration, Writing – review and editing

### Author ORCIDs
Mauna R Dasari  https://orcid.org/0000-0002-1956-2500
Susan C Alberts  https://orcid.org/0000-0002-1313-488X
Jenny Tung  https://orcid.org/0000-0003-0416-2958
Elizabeth A Archie  https://orcid.org/0000-0002-1187-0998

Reviewer #1 (Public review): https://doi.org/10.7554/eLife.102166.3.sa1
Reviewer #2 (Public review): https://doi.org/10.7554/eLife.102166.3.sa2
Author response https://doi.org/10.7554/eLife.102166.3.sa3

## Additional files

### Supplementary files
Supplementary file 1. Descriptive tables and tables reporting results from quantitative analyses. (**A**) Linear and quadratic relationships with age for alpha diversity, principal components of composition, and CLR transformed taxa present in >25% of samples (FDR threshold=0.05). (**B**) A total of 9,575 features were used to create the gut microbiome clock of aging. To be included, features must have been present in three or more samples. (**C**) Linear models of host microbiome age as predicted by host chronological age and host sex and an interaction between host age and sex. (**D**) Linear models of host microbiome age as predicted by host chronological age for each sex separately. (**E**) Predictive accuracy of several age-related metrics and traits in the Amboseli baboons. (**F**) Results of linear mixed effects models predicting microbiome delta age in female and male baboons (each sex is modeled separately). (**G**) Sources of early life adversity in the Amboseli population. (**H**) Description of developmental milestones and survival, with definitions of how censored animals were assessed. (**I**) Results of Cox proportional hazards models testing the effects of microbiome age acceleration on the time to attain three developmental milestones in female baboons (adult rank attainment, menarche, and first live birth), as well as juvenile and adult survival. (**J**) Results of Cox proportional hazards models testing the effects of microbiome age acceleration on the time to attain three developmental milestones in male baboons (testicular enlargement, natal dispersal, adult rank attainment), as well as male juvenile survival. (**K**) Comparison of four regression approaches used to construct the microbiome clock of aging. (**L**) Description of predictors used in the socio-environmental predictor and milestone analyses.

MDAR checklist

## Data availability

All data for these analyses are available on Dryad at https://doi.org/10.5061/dryad.b2rbnzspv. The 16S rRNA gene sequencing data are deposited on EBI-ENA (project ERP119849) and Qiita (study 12949). Code is available at the following GitHub repository: https://github.com/maunadasari/Dasari_etal-GutMicrobiomeAge, copy archived at *Dasari, 2025*.

The following dataset was generated:

| Author(s) | Year | Dataset title | Dataset URL | Database and Identifier |
|---|---|---|---|---|
| Dasari MR, Roche K, Jansen DA, Anderson JA, Alberts S, Tung J, Gilbert JA, Blekhman R, Mukherjee S, Archie EA | 2024 | Social and environmental predictors of gut microbiome age in wild baboons | https://doi.org/10.5061/dryad.b2rbnzspv | Dryad Digital Repository, 10.5061/dryad.b2rbnzspv |

The following previously published dataset was used:

| Author(s) | Year | Dataset title | Dataset URL | Database and Identifier |
|---|---|---|---|---|
| Grieneisen L, Dasari M, Gould TJ, Björk JR, Grenier J, Yotova V, Jansen D, Gottel N, Gordon JB, Learn NH, Gesquiere LR, Wango TL, Mututua RS, Warutere JK, Siodi L, Gilbert JA, Barreiro LB, Alberts SC | 2021 | Amboseli Baboon Research Project | https://www.ebi.ac.uk/ena/browser/view/PRJEB36635 | EBI European Nucleotide Archive, PRJEB36635 |

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

## Appendix

### Creating and assessing age-predictive machine learning models

#### Introduction to the approaches

To create our final microbiome aging clock, we tested three supervised machine learning algorithms: elastic net regression, Random Forest regression, and Gaussian process regression (*Quast et al., 2013*; *Kuznetsova et al., 2017*; *Dixon, 2003*). Below we summarize the strengths and weaknesses of each machine learning algorithm.

Elastic net regression is a regression algorithm that produces a linear model. It improves upon the predictions from simple linear regressions by incorporating coefficient penalties from the L1 regularization (LASSO regression) and L2 regularization (ridge regression) (*Kuznetsova et al., 2017*). Elastic net regression is infrequently used in microbiome studies but has produced promising results in epigenetic aging clocks due to its flexibility in choosing which features to keep and which to remove (*Horvath, 2013*; *Marioni et al., 2015*; *Chen et al., 2016*; *Binder et al., 2018*; *Anderson et al., 2021*). However, elastic net regressions produce linear relationships between the input chronological age and the predicted age, which may not accurately affect the true relationship between chronological age and the microbiome.

Random Forest regression is an ensemble learning method that creates a number of parallel decision trees, each producing its own prediction (*Dixon, 2003*). The prediction is then averaged among all trees to create the final estimate. A key advantage of Random Forest regression over elastic net regression is that it does not assume a linear relationship between the predicted estimate and the input chronological age, but the model may be biased by correlated features. Random Forest regression is commonly used in microbiome research, including other microbiome clocks (*Subramanian et al., 2014*; *Roberts, 2019*; *Benjamini and Hochberg, 1995*; *Van Rossum and Drake, 2009*).

Gaussian process regression is a nonparametric, Bayesian approach that infers a probability distribution over all the potential functions that fit the data (*Quast et al., 2013*). Like Random Forest, Gaussian process regressions do not assume a linear relationship between chronological age and predicted age but has the additional advantage of kernel customization. As such, Gaussian process regressions may be able to better handle heteroskedasticity in the data (an issue in our clock; see below). As an increase in chronological age is often associated with the breakdown of physiological processes (e.g., aging), heteroskedasticity in microbial age estimates may indicate a breakdown of the host's processes that regulate the gut microbiome.

#### Methods and optimization of machine learning algorithms

Prior to running each algorithm, all features were center log-ratio transformed within sample. We then chose a ratio of training to test dataset. To do this, we first compared the model fit of different ratios of training to test sets. These included the following training:test splits: 50:50, 60:40, 75:25, 80:20, and 90:10. We found that an 80:20 data split provided the best balance between model performance and the risk of overfitting.

In order to calculate a microbial age estimate for every sample and estimate generalization error, we used a nested cross-validation framework. Each of the three algorithms has its own internal cross-validation where a subset of the training data is held apart and used to internally validate the model. We added an additional, external layer of cross-validation with our 80:20 training:test data split. We classified samples into five different test sets where individual was as evenly represented as possible in all training and test sets. As the number of samples varied between individuals, we randomly assigned each sample a test set without replacement if an individual's sample count was less than five, or with replacement if an individual's sample count was greater than five. For each model run, four of the test datasets were treated altogether as training data and the fifth set was the validation test set.

Elastic net regressions were run in R using function `cv.glmnet()` from package glmnet (*Pedregosa et al., 2011*). The two main parameters for this model are $\lambda$, which is the penalty from the LASSO regression that penalizes extra predictors by shrinking coefficients to zero, and $\alpha$, the parameter that balances between minimizing between the residual sum of squares and minimizing the magnitude of the coefficients. cv.glmnet() automatically fits 100 values of $\lambda$ by default and names

the $\lambda$ that produces the minimum cross-validated error 'lambda.min'. We used lambda.min as our value of $\lambda$. For $\alpha$, we manually ran the model with 200 values of alpha (from 0 to 1 in increasing increments of 0.005) and picked a value of alpha that would minimize the mean absolute error and maximize the adjusted $R^2$.

Random Forest regressions were conducted in Python 3 using scikit-learn (*Callahan et al., 2016*). The main parameter was the number of decision trees being used, which defaults to 100. Too many trees could result in overfitting so in order to minimize overfitting and optimize $R^2$, we ran a series of Random Forest regressions with different numbers of trees: we increased the number of trees in increments of 50, stopping at 400 because of minimal changes in $R^2$ relative to 200 trees.

Gaussian process regressions were also conducted in Python 3 using scikit-learn (*Callahan et al., 2016*; *Wright et al., 2012*). In both the non-heteroskedastic-kernel model and heteroskedastic-kernel model, the main parameters we used to modify the kernel function included the scale and bounds. These parameters moderate the level of overfitting in the algorithm: the scale parameter specifies a starting point for which the algorithm optimizes within the confines of the bounds parameters. As with the other models, we incrementally changed both the scale parameter within a wide range of bounds and checked the output model's $R^2$ and median error. Our final model retained a wide range of bounds (1–100) and set the scale parameter to the median euclidian distance of the dataset as calculated in R using function `vegdist()` from R package vegan (*Sender et al., 2016*).

Due to the heteroskedasticity exhibited by the models above (*Figure 3—figure supplement 2*), we modified the Gaussian process regression's kernel function further to account for the variance within the dataset. Specifically, we multiplied the variance in the training data by the radial basis function, which distributed the higher variance in later life more evenly across lifespan.

## Comparison of machine learning algorithms

To assess model accuracy, we used the predicted age estimates from all five runs of the nested cross-validation procedure to assess model fit and accuracy. As in *Horvath, 2013*, we regressed the sample's predicted microbial age ($age_m$) against the host's known chronological age ($age_c$) and calculated: (1) the $R^2$ between $age_c$ and $age_m$; (2) the Pearson's correlation coefficient between $age_c$ and $age_m$; and (3) the median error as the median absolute difference between $age_c$ and $age_m$ (*Supplementary file 1K* and *Figure 3—figure supplement 1*). Across all algorithms, we observed that males always aged faster than females, which is consistent with well-known patterns of sex-specific senescence in humans and other primates (*Gloor et al., 2017*; *Figure 3—figure supplement 1*). The Gaussian process regression with the heteroskedastic kernel was the best model for every metric assessed—it maximized $R^2$ and Pearson's $R$ to 0.488 and 0.698 (respectively) while minimizing median error. It also was the only model with which we were able to alleviate any heteroskedasticity.

