## [Editor Report · eLife Assessment]

This study leverages an impressive and comprehensive longitudinal 16S rRNA gut microbiome dataset from baboons to provide **important** insight regarding the use of a microbiome-based clock to predict biological age. The evidence for age-associated microbiome features and environmental and social variables that impact microbiome aging is **convincing**. This study of microbiomes as markers of host age will fuel inquiries and studies and interest a broad range of researchers, especially those interested in alternatives to measuring biological aging.

---

## [Referee Report · Reviewer #1 (Public review)]

Summary:

The authors used a subset of a very large, previously generated 16S dataset to: (1) assess age-associated features; and (2) develop a fecal microbiome clock, based on extensive longitudinal sampling of wild baboons for which near-exact chronological age is known. They further seek to understand deviation from age-expected patterns and uncover if and why some individuals have an older or younger microbiome than expected, and the health and longevity implications of such variation. Overall, the authors compellingly achieved their goals to discover age-associated microbiome features and develop a fecal microbiome clock. They also showed clear and exciting evidence for sex and rank-associated variation in the pace of gut microbiome aging and impacts of seasonality on microbiome age in females. These data add to a growing understanding of modifiers of the pace of age in primates, and links among different biological indicators of age, with implications for understanding and contextualizing human variation. However, in the current version there are gaps in the analyses with respect to the social environment, and in comparisons with other biological indicators of age. Despite this, I anticipate this work will be impactful, generate new areas of inquiry and fuel additional comparative studies.

Strengths:

The major strengths of the paper are the size and sampling depth of the study population, including ability to characterize of the social and physical environments, and the application of recent and exciting methods to characterize the microbiome clock. An additional strength was the ability of the authors to compare and contrast the relative age-predictive power of the fecal microbiome clock to other biological methods of age estimation available for the study population (dental wear, blood cell parameters, methylation data). Furthermore, the writing and support materials are clear and informative and visually appealing.

Revisions made following initial review have further improved the content and clarity.

Weaknesses:

Revisions to the manuscript clarified some of the analysis decisions and limitations regarding drawing comparisons between the microbiome clock and other metrics of biological age, and on the impact of sociality on microbiome metrics. Hopefully these interesting topics will be further addressed in forthcoming publications.

---

## [Referee Report · Reviewer #2 (Public review)]

Summary:

Dasari et al present an interesting study investigating the use of 'microbiota age' as an alternative to other measures of 'biological age'. The study provides several curious insights into biological ageing. Although 'microbiota age' holds potential as a proxy of biological age, it comes with limitations considering the gut microbial community can be influenced various non-age related factors, and various age-related stressors may not manifest in changes in the gut microbiota.

Strengths:

The dataset this study is based on is impressive, and can reveal various insights into biological ageing and beyond. The analysis implemented is extensive and of high level.

Weaknesses:

The key weakness is the use of microbiota age instead of e.g., DNA-methylation based epigenetic age as a proxy of biological ageing, for reasons stated in the summary. DNA methylation levels can be measured from faecal samples, and as such epigenetic clocks too can be non-invasive.

In the first round of review, I provided authors a list of minor edits, which they have implemented in the revised version of the manuscript.

---

## [Author Response]

The following is the authors’ response to the original reviews.

**Public Reviews:**

**Reviewer #1 (Public review):**
Summary:The authors used a subset of a very large, previously generated 16S dataset to:(1) Assess age-associated features; and (2) develop a fecal microbiome clock, based on an extensive longitudinal sampling of wild baboons for which near-exact chronological age is known. They further seek to understand deviation from age-expected patterns and uncover if and why some individuals have an older or younger microbiome than expected, and the health and longevity implications of such variation. Overall, the authors compellingly achieved their goals of discovering age-associated microbiome features and developing a fecal microbiome clock. They also showed clear and exciting evidence for sex and rank-associated variation in the pace of gut microbiome aging and impacts of seasonality on microbiome age in females. These data add to a growing understanding of modifiers of the pace of age in primates, and links among different biological indicators of age, with implications for understanding and contextualizing human variation. However, in the current version, there are gaps in the analyses with respect to the social environment, and in comparisons with other biological indicators of age. Despite this, I anticipate this work will be impactful, generate new areas of inquiry, and fuel additional comparative studies.

Thank you for the supportive comments and constructive reviews.

Strengths:The major strengths of the paper are the size and sampling depth of the study population, including the ability to characterize the social and physical environments, and the application of recent and exciting methods to characterize the microbiome clock. An additional strength was the ability of the authors to compare and contrast the relative age-predictive power of the fecal microbiome clock to other biological methods of age estimation available for the study population (dental wear, blood cell parameters, methylation data). Furthermore, the writing and support materials are clear, informative and visually appealing.Weaknesses:It seems clear that more could be done in the area of drawing comparisons among the microbiome clock and other metrics of biological age, given the extensive data available for the study population. It was confusing to see this goal (i.e. "(i) to test whether microbiome age is correlated with other hallmarks of biological age in this population"), listed as a future direction, when the authors began this process here and have the data to do more; it would add to the impact of the paper to see this more extensively developed.

Comparing the microbiome clock to other metrics of biological age in our population is a high priority (these other metrics of biological age are in Table S5 and include epigenetic age measured in blood, the non-invasive physiology and behavior clock (NPB clock), dentine exposure, body mass index, and blood cell counts (Galbany et al. 2011; Altmann et al. 2010; Jayashankar et al. 2003; Weibel et al. 2024; Anderson et al. 2021)). However, we have opted to test these relationships in a separate manuscript. We made this decision because of the complexity of the analytical task: these metrics were not necessarily collected on the same subjects, and when they were, each metric was often measured at a different age for a given animal. Further, two of the metrics (microbiome clock and NPB clock) are measured longitudinally within subjects but on different time scales (the NPB clock is measured annually while microbiome age is measured in individual samples). The other metrics are cross-sectional. Testing the correlations between them will require exploration of how subject inclusion and time scale affect the relationships between metrics.

We now explain the complexity of this analysis in the discussion in lines 447-450. In addition, we have added the NPB clock (Weibel et al. 2024) to the text in lines 260-262 and to Table S5.

An additional weakness of the current set of analyses is that the authors did not explore the impact of current social network connectedness on microbiome parameters, despite the landmark finding from members of this authorship studying the same population that "Social networks predict gut microbiome composition in wild baboons" published here in eLife some years ago. While a mother's social connectedness is included as a parameter of early life adversity, overall the authors focus strongly on social dominance rank, without discussion of that parameter's impact on social network size or directly assessing it.

Thank you for raising this important point, which was not well explained in our manuscript. We find that the signatures of social group membership and social network proximity are only detectable our population for samples collected close in time. All of the samples analyzed in Tung et al. 2015 (“Social networks predict gut microbiome composition in wild baboons”) were collected within six weeks of each other. By contrast, the data set analyzed here spans 14 years, with very few samples from close social partners collected close in time. Hence, the effects of social group membership and social proximity are weak or undetectable. We described these findings in Grieneisen et al. 2021 and Bjork et al. 2022, and we now explain this logic on line 530, which states, “We did not model individual social network position because prior analyses of this data set find no evidence that close social partners have more similar gut microbiomes, probably because we lack samples from close social partners sampled close in time (Grieneisen et al. 2021; Björk et al. 2022).”

We do find small effects of social group membership, which is included as a random effect in our models of how each microbiome feature is associated with host age (line 529) and our models predicting microbiome Dage (line 606; Table S6).

**Reviewer #2 (Public review):**
Summary:Dasari et al present an interesting study investigating the use of 'microbiota age' as an alternative to other measures of 'biological age'. The study provides several curious insights into biological aging. Although 'microbiota age' holds potential as a proxy of biological age, it comes with limitations considering the gut microbial community can be influenced by various non-age related factors, and various age-related stressors may not manifest in changes in the gut microbiota. The work would benefit from a more comprehensive discussion, that includes the limitations of the study and what these mean to the interpretation of the results.

We agree and have text to the discussion that expands on the limitations of this study and what those limitations mean for the interpretation of the results. For instance, lines 395-400 read, “Despite the relative accuracy of the baboon microbiome clock compared to similar clocks in humans, our clock has several limitations. First, the clock’s ability to predict individual age is lower than for age clocks based on patterns of DNA methylation—both for humans and baboons (Horvath 2013; Marioni et al. 2015; Chen et al. 2016; Binder et al. 2018; Anderson et al. 2021). One reason for this difference may be that gut microbiomes can be influenced by several non-age-related factors, including social group membership, seasonal changes in resource use, and fluctuations in microbial communities in the environment”

In addition, lines 405-411 now reads, “Third, the relationships between potential socio-environmental drivers of biological aging and the resulting biological age predictions were inconsistent. For instance, some sources of early life adversity were linked to old-for-age gut microbiomes (e.g., males born into large social groups), while others were linked to young-for-age microbiomes (e.g., males who experienced maternal social isolation or early life drought), or were unrelated to gut microbiome age (e.g., males who experienced maternal loss; any source of early life adversity in females).”

Strengths:The dataset this study is based on is impressive, and can reveal various insights into biological ageing and beyond. The analysis implemented is extensive and high-level.Weaknesses:The key weakness is the use of microbiota age instead of e.g., DNA-methylation-based epigenetic age as a proxy of biological ageing, for reasons stated in the summary. DNA methylation levels can be measured from faecal samples, and as such epigenetic clocks too can be non-invasive. I will provide authors a list of minor edits to improve the read, to provide more details on Methods, and to make sure study limitations are discussed comprehensively.

Thank you for this point. In response, we have deleted the text from the discussion that stated that non-invasive sampling is an advantage of microbiome clocks. In addition, we now propose a non-invasive epigenetic clock from fecal samples as an important future direction for our population (see line 450).

**Recommendations for the authors:**

**Reviewer #1 (Recommendations for the authors):**
Abstract - The opening 2 sentences are not especially original or reflective of the potential value/ premise of the study. Members of this team have themselves measured variation in biological age in many different ways, and the implication that measuring a microbiome clock is easy or straightforward is not compelling. This paper is very interesting and provides unique insight, but I think overall there is a missed opportunity in the abstract to emphasize this, given the innovative science presented here. Furthermore, the last 2 sentences of the abstract are especially interesting - but missing a final statement on the broader significance of research outside of baboons.

We appreciate these comments and have revised the Abstract accordingly. The introductory sentences now read, “Mammalian gut microbiomes are highly dynamic communities that shape and are shaped by host aging, including age-related changes to host immunity, metabolism, and behavior. As such, gut microbial composition may provide valuable information on host biological age.” (lines 31-34). The last two sentences of the abstract now read, “Hence, in our host population, gut microbiome age largely reflects current, as opposed to past, social and environmental conditions, and does not predict the pace of host development or host mortality risk. We add to a growing understanding of how age is reflected in different host phenotypes and what forces modify biological age in primates.” (lines 40-43).

If possible, it would be highly useful to present some comments on concordance in patterns at different levels. Are all ASVs assessed at both the family and genus levels? Do they follow similar patterns when assessed at different levels? What can we learn about the system by looking at different levels of taxonomic assignment?

The section on relationships between host age and individual microbiome features is already lengthy, so we have not added an analysis of concordance between different taxonomic levels. However, we added a justification for why we tested for age signatures in different levels of taxa to line 171, which reads, “We tested these different taxonomic levels in order to learn whether the degree to which coarse and fine-grained designations categories were associated with host age.”

To calculate the delta age - please clarify if this was done at the level of years, as suggested in Figure 3C, or at the level of months or portion months, etc?

Delta age is measured in years. This is now clarified in lines 294, 295, and 578.

Spelling mistake in table S12, cell B4 (Octovber)

Thank you. This typo has been corrected.

Given the start intro with vertebrates, the second paragraph needs some tweaking to be appropriate. Perhaps, "At least among mammals, one valuable marker of biological aging may lie in the composition and dynamics of the mammalian gut microbiome (7-10)." Or simply remove "mammalian".

We have updated this sentence based on your suggestions in line 54. It reads, “In mammals, one valuable marker of biological aging may lie in the composition and dynamics of the gut microbiome (Claesson et al. 2012; Heintz and Mair 2014; O’Toole and Jeffery 2015; Sadoughi et al. 2022).”

A rewrite at the end of the introduction is needed to avoid the almost direct repetition in lines 115-118 and 129-131 (including lit cited). One potentially effective way to approach this is to keep the predictions in the earlier paragraph and then more clearly center the approach and the overarching results statement in the latter paragraph. (I.e., "we find that season and social rank have stronger effects on microbiome age than early life events. Further, microbiome age does not predict host development or mortality.").

Thank you for pointing this out. We have re-organized the predictions in the introduction based on your suggestion. The alternative “recency effects” model now appears in the paragraph that starts in line 110. The final paragraph then centers on the overall approach and the results statement (lines 128-140)

Be clear in each case where taxon-level trends are discussed if it's at Family, Genus, or other level. It's there most, but not all, of the time.

We have gone through the text and clarified what taxa or microbiome feature was the subject of our analyses in any places where this was not clear.

In the legend for Figure 2, add clarification for how values to right versus left of the centered value should be interpreted with respect to age (e.g. "values to x of the center are more abundant in older individuals").

We now clarify in Figure 2C and 2D that “Positive values are more abundant in older hosts”.

Figure 3 - Are Panels A, B, and C all needed - can the value for all individuals not also be overlaid in the panel showing sex differences and the same point showing individuals with "old" and "young" microbiomes be added in the same plot if it was slightly larger?

We agree and have simplified Figure 3. We reduced the number of panels from three to two, and we added the information about how to calculate delta age to Panel A. We also moved the equation from the top of Panel C to the bottom right of Panel A.

**Reviewer #2 (Recommendations for the authors):**
Dasari et al present an interesting study investigating the use of 'microbiota age' as an alternative to other measures of 'biological age'. The study provides several curious insights which in principle warrant publication. However, I do think the manuscript should be carefully revised. Below I list some minor revisions that should be implemented. Importantly, the authors should discuss in the Discussion the pros and cons of using 'microbiota age' as a proxy of 'biological age'. Further, the authors should provide more information on Methods, to make sure the study can be replicated.

Thank you for these important points. Based on your comments and those of the first reviewer, we have expanded our discussion of the limitations of using microbiota age as a proxy for biological age (see edits to the paragraph starting in line 395).

We have also expanded our methods around sample collection, DNA extraction, and sequencing to describe our sampling methods, strategies to mitigate and address possible contamination, and batch effects. See lines 483-490 and our citations to the original papers where these methods are described in detail.

(1) Lines 85-99: I think this paragraph could be revisited to make the assumptions clearer. For instance, the last sentence is currently a little confusing: are authors expecting males to exhibit old-for-age microbiomes already during the juvenile period?

This prediction has been clarified. Line 96 now reads, “Hence, we predicted that adult male baboons would exhibit gut microbiomes that are old-for-age, compared to adult females (by contrast, we expected no sex effects on microbiome age in juvenile baboons).”

(2) Lines 118-121: Could the authors discuss this assumption in relation to what has been observed e.g., in humans in terms of delays in gut microbiome development? Delayed/accelerated gut microbiome development has been studied before, so this assumption would be stronger if related to what we know from previous studies.

This comment refers to the sentence which originally stated, “However, we also expected that some sources of early life adversity might be linked to young-for-age gut microbiota. For instance, maternal social isolation might delay gut microbiome development due to less frequent microbial exposures from conspecifics.” We have slightly expanded the text here (line 117) to explain our logic. We now include citations for our predictions. We did not include a detailed discussion of prior literature on microbiome development in the interest of keeping the same level of detail across all sections on our predictions.

(3) As the authors discuss, various adversities can lead to old-for-age but also young-for-age microbiome composition. This should be discussed in the limitations.

We agree. This is now discussed in the sentence starting at line 371, which reads, “…deviations from microbiome age predictions are explained by socio-environmental conditions experienced by individual hosts, especially recent conditions, although the effect sizes are small and are not always directionally consistent.” In addition, the text starting at line 405 now reads, “Third, the relationships between potential socio-environmental drivers of biological aging and the resulting biological age predictions were inconsistent. For instance, some sources of early life adversity were linked to old-for-age gut microbiomes (e.g., males born into large social groups), while others were linked to young-for-age microbiomes (e.g., males who experienced maternal social isolation or early life drought), or were unrelated to gut microbiome age (e.g., males who experienced maternal loss; any source of early life adversity in females).”

(4) In various places, e.g., lines 129-131, it is a little unclear at what chronological age authors are expecting microbiota to appear young/old-for-age.

This sentence was removed while responding to the comments from the first reviewer.

(5) Lines 132-133: this statement could be backed by stating that this is because the gut microbiota can change rapidly e.g., when diet changes (or whatever the authors think could be behind this).

We have added an expository sentence at line 123, including new citations. This sentence reads, “Indeed, gut microbiomes are highly dynamic and can change rapidly in response to host diet or other aspects of host physiology, behavior, or environments”.

We now cite:

· Hicks, A.L., et al. (2018). Gut microbiomes of wild great apes fluctuate seasonally in response to diet. Nature Communications 9, 1786.

· Kolodny, O., et al. (2019). Coordinated change at the colony level in fruit bat fur microbiomes through time. Nature Ecology & Evolution 3, 116-124.

· Risely, A., et al. (2021) Diurnal oscillations in gut bacterial load and composition eclipse seasonal and lifetime dynamics in wild meerkats. Nat Commun 12, 6017.

(6) Lines 135-137: current or past season and social rank? This paragraph introduces the idea that it could be past rather than current socio-environmental factors that might predict microbiota age, so the authors should clarify this sentence.

We have clarified the information in this sentence. line 135 now reads, “In general, our results support the idea that a baboon’s current socio-environmental conditions, especially their current social rank and the season of sampling, have stronger effects on microbiome age than early life events—many of which occurred many years prior to sampling.”

(7) Lines 136-137: this sentence could include some kind of a conclusion of this finding. What might this mean?

We have added a sentence at line 138, which speculates that, “…the dynamism of the gut microbiome may often overwhelm and erase early life effects on gut microbiome age.”

(8) Use 'microbiota' or 'microbiome' across the manuscript; currently, the terms are used interchangeably. I don't have a strong opinion on this, although typically 'microbiota' is used when data comes from 16S rRNA.

We have updated the text to replace any instance of “microbiota” with “microbiome”. We use the term microbiome in the sense of this definition from the National Human Genome Research Institute, which defines a microbiome as “the community of microorganisms (such as fungi, bacteria and viruses) that exists in a particular environment”.

(9) Figure 1 legend: make sure to unify formatting; e.g., present sample sizes as N = or n = , rather than both, and either include or do not include commas in 4-digit values (sample sizes).

We have checked the formatting related to sample sizes and the use of commas in 4-digits in the main text and supplement. The formats are now consistent.

(10) Line 166: relative abundances surely?

Following Gloor et al. (2017), our analyses use centered log-ratio (CLR) transformations of read counts, which is the recommended approach for compositional data such as 16S rRNA amplicon read counts. CLR transformations are scale-invariant, so the same ratio is obtained in a sample with few read versus many reads. We now cite Gloor et al. (2017) at line 169 and in the methods in line 517, which reads “centered log ratio (CLR) transformed abundances (i.e., read counts) of each microbial phyla (n=30), family (n=290), genus (n=747), and amplicon sequence variance (ASV) detected in >25% of samples (n=358). CLR transformations are a recommended approach for addressing the compositional nature of 16S rRNA amplicon read count data (Gloor et al. 2017).”

(11) Lines 167-172: were technical factors, e.g., read depth or sequencing batch, included as random effects?

Thank you for catching this oversight in the text. We did model sequencing depth and batch effects. The sentence starting at line 173 now reads, “For each of these 1,440 features, we tested its association with host age by running linear mixed effects models that included linear and quadratic effects of host age and four other fixed effects: sequencing depth, the season of sample collection (wet or dry), the average maximum temperature for the month prior to sample collection, and the total rainfall in the month prior to sample collection (Grieneisen et al. 2021; Björk et al. 2022; Tung et al. 2015). Baboon identity, social group membership, hydrological year of sampling, and sequencing plate (as a batch effect) were modeled as random effects.”

(12) Lines 175-180: When discussing how these alpha diversity results relate to previous findings, the authors should be clear about whether they talk about weighted or non-weighted measures of alpha diversity. - also maybe this should be included in the discussion rather than the results? Please consider this when revisiting the manuscript (see how it reads after edits).

Richness is the only unweighted metric, which we now clarify in line 181. We opted to retain the interpretation in the text in its original location to maintain the emphasis in the discussion on the microbiome clock results.

(13) Table S1 is very hard to interpret in the provided PDF format as columns are not presented side-by-side. It is currently hard to check model output for e.g., specific families. This needs to be revisited.

We agree. We believe that eLife’s submission portal automatically generates a PDF for any supplementary item. However, we also include the supplementary tables as an Excel workbook which has the columns presented side-by-side.

(14) Line 184: taxa meaning what? Unclear what authors refer to with this sentence, taxa across taxonomic levels, or ASVs, or what does the 51.6% refer to?

We have edited line 191 to clarify that this sentence refers to taxa at all taxonomic levels (phyla to ASVs).

(15) Line 191: a punctuation mark missing after ref (81).

We have added the missing period at the end of this sentence.

(16) Lines 189-197: this should go into the discussion in my opinion.

We have opted to retain this interpretation, now at line 183.

(17) Lines 215-219: Not sure what this means; do the authors mean features were not restricted to age-associated taxa, ie also e.g., diversity and other taxa-independent patterns were included? If so, the rest of the highlighted lines should be revisited to make this clear, currently to me it is very unclear what 'These could include features that are not strongly age-correlated in isolation' means. Currently, that sounds like some features included were only age-associated in combination with other features, but unclear how this relates to taxa-dependency/taxa-independency.

We agree this was not clear. We have revised line 224 to read, “We included all 9,575 microbiome features in our age predictions, as opposed to just those that were statistically significantly associated with age because removing these non-significant features could exclude features that contribute to age prediction via interactions with other taxa.”

(18) Line 403-407: There is now a paper showing epigenetic clocks can be built with faecal samples, so this argument is not valid. Please revisit in light of this publication: https://onlinelibrary.wiley.com/doi/epdf/10.1111/mec.17330

Thank you for bringing this paper to our attention. We deleted the text that describes epigenetic clocks as invasive, and we now cite this paper in line 450, which reads, “We also hope to measure epigenetic age in fecal samples, leveraging methods developed in Hanski et al. 2024.”

(19) Line 427: a punctuation mark/semicolon missing before However.

We have corrected this typo.

(20) Lines 419-428: I don't quite understand this speculation. Why would the priority of access to food lead to an old-looking gut microbiome? This paragraph needs stronger arguments, currently unclear and also not super convincing.

We agree this was confusing. We have revised this text to clarify the explanation. The text starting at line 424 now reads, “This outcome points towards a shared driver of high social status in shaping gut microbiome age in both males and females. While it is difficult to identify a plausible shared driver, one benefit shared by both high-ranking males and females is priority of access to food. This access may result in fewer foraging disruptions and a higher quality, more stable diet. At the same time, prior research in Amboseli suggests that as animals age, their diets become more canalized and less variable (Grieneisen et al. 2021). Hence aging and priority of access to food might both be associated with dietary stability and old-for-age microbiomes. However, this explanation is speculative and more work is needed to understand the relationship between rank and microbiome age.”

(21) Line 434: remove 'be'.

We have corrected this typo.

(22) Line 478: add information on how samples were collected; e.g., were samples collected from the ground? How was cross-contamination with soil microbiota minimised? Were samples taken from the inner part of depositions? These factors can influence microbiota samples quite drastically so detailed info is needed. Also what does homogenisation mean in this context? How soon were samples freeze-dried after sample collection?

We have expanded our methods with respect to sample collection. This text starts in line 483 and reads, “Samples were collected from the ground within 15 minutes of defecation. For each sample, approximately 20 g of feces was collected into a paper cup, homogenized by stirring with a wooden tongue depressor, and a 5 g aliquot of the homogenized sample was transferred to a tube containing 95% ethanol. While a small amount of soil was typically present on the outside of the fecal sample, mammalian feces contains 1000 times the number of microbial cells in a typical soil sample (Sender, Fuchs, and Milo 2016; Raynaud and Nunan 2014), which overwhelms the signal of soil bacteria in our analyses (Grieneisen et al. 2021). Samples were transported from the field in Amboseli to a lab in Nairobi, freeze-dried, and then sifted to remove plant matter prior to long term storage at -80°C.”

(23) Line 480 onwards: were negative controls included in extraction batches? Were samples randomised into extraction batches?

Yes, we included extraction blanks. These are now described in lines 495-500. This text reads, “We included one extraction blank per batch, which had significantly lower DNA concentrations than sample wells (t-test; t=-50, p < 2.2x10-16; Grieneisen et al. 2021). We also included technical replicates, which were the same fecal sample sequenced across multiple extraction and library preparation batches. Technical replicates from different batches clustered with each other rather than with their batch, indicating that true biological differences between samples are larger than batch effects.”

(24) Were extraction, library prep, and sequencing negative controls included? Is data available?

We included extraction blanks (described above) and technical replicates, which were the same sample sequenced across multiple extraction and library preparation batches. Technical replicates from different batches clustered with each other rather than with their batch, indicating that true biological differences between samples are larger than batch effects.

We have updated the data availability statement to read, “All data for these analyses are available on Dryad at https://doi.org/10.5061/dryad.b2rbnzspv. The 16S rRNA gene sequencing data are deposited on EBI-ENA (project ERP119849) and Qiita (study 12949). Code is available at the following GitHub repository: https://github.com/maunadasari/Dasari_etal-GutMicrobiomeAge”.

(25) Line 562: how were corrected microbiome delta ages calculated? Currently, the authors state x, y and z factors were corrected for, but it is unclear how this was done.

The paragraph starting at line 577 describes how microbiome delta age was calculated. We have made only a few changes to this text because we were not sure which aspects of these methods confused the reviewer. However, briefly, we calculated sample-specific microbiome Dage in years as the difference between a sample’s microbial age estimate, age_m_ from the microbiome clock, and the host’s chronological age in years at the time of sample collection, age_c_. Higher microbiome Dages indicate old-for-age microbiomes, as age_m_ > age_c_, and lower values (which are often negative) indicate a young-for-age microbiome, where age_c_ > age_m_ (see Figure 3).

(26) Line 579: typo 'as'.

We have corrected this typo.

Works Cited

Altmann, Jeanne, Laurence Gesquiere, Jordi Galbany, Patrick O Onyango, and Susan C Alberts. 2010. “Life History Context of Reproductive Aging in a Wild Primate Model.” Annals of the New York Academy of Sciences 1204:127–38. https://doi.org/10.1111/j.1749-6632.2010.05531.x.

Anderson, Jordan A, Rachel A Johnston, Amanda J Lea, Fernando A Campos, Tawni N Voyles, Mercy Y Akinyi, Susan C Alberts, Elizabeth A Archie, and Jenny Tung. 2021. “High Social Status Males Experience Accelerated Epigenetic Aging in Wild Baboons.” Edited by George H Perry. eLife 10 (April):e66128. https://doi.org/10.7554/eLife.66128.

Binder, Alexandra M., Camila Corvalan, Verónica Mericq, Ana Pereira, José Luis Santos, Steve Horvath, John Shepherd, and Karin B. Michels. 2018. “Faster Ticking Rate of the Epigenetic Clock Is Associated with Faster Pubertal Development in Girls.” Epigenetics 13 (1): 85–94. https://doi.org/10.1080/15592294.2017.1414127.

Björk, Johannes R., Mauna R. Dasari, Kim Roche, Laura Grieneisen, Trevor J. Gould, Jean-Christophe Grenier, Vania Yotova, et al. 2022. “Synchrony and Idiosyncrasy in the Gut Microbiome of Wild Baboons.” Nature Ecology & Evolution, June, 1–10. https://doi.org/10.1038/s41559-022-01773-4.

Chen, Brian H., Riccardo E. Marioni, Elena Colicino, Marjolein J. Peters, Cavin K. Ward-Caviness, Pei-Chien Tsai, Nicholas S. Roetker, et al. 2016. “DNA Methylation-Based Measures of Biological Age: Meta-Analysis Predicting Time to Death.” Aging (Albany NY) 8 (9): 1844–59. https://doi.org/10.18632/aging.101020.

Claesson, Marcus J., Ian B. Jeffery, Susana Conde, Susan E. Power, Eibhlís M. O’Connor, Siobhán Cusack, Hugh M. B. Harris, et al. 2012. “Gut Microbiota Composition Correlates with Diet and Health in the Elderly.” Nature 488 (7410): 178–84. https://doi.org/10.1038/nature11319.

Galbany, Jordi, Jeanne Altmann, Alejandro Pérez-Pérez, and Susan C. Alberts. 2011. “Age and Individual Foraging Behavior Predict Tooth Wear in Amboseli Baboons.” American Journal of Physical Anthropology 144 (1): 51–59. https://doi.org/10.1002/ajpa.21368.

Gloor, Gregory B., Jean M. Macklaim, Vera Pawlowsky-Glahn, and Juan J. Egozcue. 2017. “Microbiome Datasets Are Compositional: And This Is Not Optional.” Frontiers in Microbiology 8. https://doi.org/10.3389/fmicb.2017.02224.

Grieneisen, Laura E., Mauna Dasari, Trevor J. Gould, Johannes R. Björk, Jean-Christophe Grenier, Vania Yotova, David Jansen, et al. 2021. “Gut Microbiome Heritability Is Nearly Universal but Environmentally Contingent.” Science 373 (6551): 181–86. https://doi.org/10.1126/science.aba5483.

Hanski, Eveliina, Susan Joseph, Aura Raulo, Klara M. Wanelik, Áine O’Toole, Sarah C. L. Knowles, and Tom J. Little. 2024. “Epigenetic Age Estimation of Wild Mice Using Faecal Samples.” Molecular Ecology 33 (8): e17330. https://doi.org/10.1111/mec.17330.

Heintz, Caroline, and William Mair. 2014. “You Are What You Host: Microbiome Modulation of the Aging Process.” Cell 156 (3): 408–11. http://dx.doi.org/10.1016/j.cell.2014.01.025.

Horvath, Steve. 2013. “DNA Methylation Age of Human Tissues and Cell Types.” Genome Biology 14 (10): R115. https://doi.org/10.1186/gb-2013-14-10-r115.

Jayashankar, Lakshmi, Kathleen M. Brasky, John A. Ward, and Roberta Attanasio. 2003. “Lymphocyte Modulation in a Baboon Model of Immunosenescence.” Clinical and Vaccine Immunology 10 (5): 870–75. https://doi.org/10.1128/CDLI.10.5.870-875.2003.

Marioni, Riccardo E., Sonia Shah, Allan F. McRae, Brian H. Chen, Elena Colicino, Sarah E. Harris, Jude Gibson, et al. 2015. “DNA Methylation Age of Blood Predicts All-Cause Mortality in Later Life.” Genome Biology 16 (1): 25. https://doi.org/10.1186/s13059-015-0584-6.

O’Toole, Paul W., and Ian B. Jeffery. 2015. “Gut Microbiota and Aging.” Science 350 (6265): 1214–15. https://doi.org/10.1126/science.aac8469.

Raynaud, Xavier, and Naoise Nunan. 2014. “Spatial Ecology of Bacteria at the Microscale in Soil.” PLOS ONE 9 (1): e87217. https://doi.org/10.1371/journal.pone.0087217.

Sadoughi, Baptiste, Dominik Schneider, Rolf Daniel, Oliver Schülke, and Julia Ostner. 2022. “Aging Gut Microbiota of Wild Macaques Are Equally Diverse, Less Stable, but Progressively Personalized.” Microbiome 10 (1): 95. https://doi.org/10.1186/s40168-022-01283-2.

Sender, Ron, Shai Fuchs, and Ron Milo. 2016. “Revised Estimates for the Number of Human and Bacteria Cells in the Body.” PLOS Biology 14 (8): e1002533. https://doi.org/10.1371/journal.pbio.1002533.

Tung, J, L B Barreiro, M B Burns, J C Grenier, J Lynch, L E Grieneisen, J Altmann, S C Alberts, R Blekhman, and E A Archie. 2015. “Social Networks Predict Gut Microbiome Composition in Wild Baboons.” Elife 4. https://doi.org/10.7554/eLife.05224.

Weibel, Chelsea J., Mauna R. Dasari, David A. Jansen, Laurence R. Gesquiere, Raphael S. Mututua, J. Kinyua Warutere, Long’ida I. Siodi, Susan C. Alberts, Jenny Tung, and Elizabeth A. Archie. 2024. “Using Non-Invasive Behavioral and Physiological Data to Measure Biological Age in Wild Baboons.” GeroScience 46 (5): 4059–74. https://doi.org/10.1007/s11357-024-01157-5.